# Genome-resolved biogeography of Phaeocystales, cosmopolitan bloom-forming algae

Zoltán Füssy [1,2,3], Robert H. Lampe [1,2], Kevin R. Arrigo [4], Kerrie Barry [5], Margaret M. Brisbin[6], Corina P. D. Brussaard [7,8], Johan Decelle [9], Colomban de Vargas[10], Giacomo R. DiTullio[11], Liam D. H. Elbourne [12,13], Marc E. Frischer[14], David M. Goodstein[5], Igor V. Grigoriev [5,15], Richard D. Hayes [5], Adam L. Healey[16], Chase C. James [17], Jerry W. Jenkins [16], Caroline Juery[9,28], Manish Kumar [18], Adam B. Kustka [19], Florian Maumus [20,29], Anna M. G. Novák Vanclová[21], Miroslav Oborník[3,22], Ian T. Paulsen [12,13], Ian Probert [10], Mak A. Saito [23], Jeremy Schmutz [5,16], Tomáš Skalický[22], Diego Tec-Campos[18], Hannah Tomelka[20,24], Pavlína Věchtová[3], Pratap Venepally[1,2], Brendan Wilson-Mortier[12], Karsten Zengler [18,25,26,27], Hong Zheng[2] & Andrew E. Allen [1,2] ✉

Phaeocystales, comprising the genus *Phaeocystis* and an uncharacterized sister lineage, are nanoplanktonic haptophytes widespread in the global ocean. Several species form mucilaginous colonies and influence key biogeochemical cycles, yet their underlying diversity and ecological strategies remain underexplored. Here, we present new genomic data from 13 strains, including three high-quality reference genomes (N50 > 30 kbp), and integrate previous metagenome-assembled genomes to resolve a robust phylogeny. Divergence timing of *P. antarctica* aligns with Miocene cooling and Southern Ocean isolation. Genomic traits reveal metabolic flexibility, including mixotrophic nitrogen acquisition in temperate waters and gene expansions linked to polar nutrient adaptation. Concordantly, transcriptomic comparisons between temperate and polar *Phaeocystis* suggest Southern Ocean populations experience iron and $B_{12}$ limitation. We also identify signatures of horizontal gene transfer and endogenous giant virus/virophage insertions. Together, these findings highlight Phaeocystales as an ecologically versatile and geographically widespread lineage shaped by evolutionary innovation and adaptation to contrasting environmental stressors.

*Phaeocystis* (Haptophyta) are ecologically versatile algae occurring in virtually all photic marine environments[1–3]. As keystone phytoplankton that shape the structure and functions of marine ecosystems, *Phaeocystis* is the only impactful algal genus recognized as a distinct phytoplankton functional type (PFT)[4,5], or so-called trophic engineer[6].

*Phaeocystis* have a profound effect on the global circulation of organic carbon[2,7,8] and sulfur[9], and often form seasonal high-biomass blooms[10,11]. They can account for 4.3-10.1% of global plankton biomass[5,12] and approximately 2-4% of marine eukaryotic rDNA[13,14] (Fig. 1a). With primary production estimated at >1 g C m$^{-2}$day$^{-1}$ [15],

**Fig. 1 | Significance of *Phaeocystis* spp. a** Global abundance of *Phaeocystis* is in the range of 1–28 % of total marine eukaryotes, on par with well-known algal groups (e.g., diatoms and coccolithophores) and zooplankton. These figures highlight their importance in marine environments as primary producers. Estimates are based on biomass (MAREDAT), total occurrence in unigene collections, and based on genome-mapping of environmental reads (this study). Box-and-whisker plots within the violin plots here and in **b** show median, interquartile range, and 1.5*IQR values. **b** Environmental expression of organosulfur compound (DMSP and DMS) biosynthetic genes shows *Phaeocystis*-specific expression in comparison to other eukaryotic groups. Data from MATOU. **c** Generalized life cycle of colony-forming *Phaeocystis* (*P. antarctica*, *P. globosa*, and *P. pouchetii*) with four main morphotypes; two types of scale-forming haploid flagellates; a diploid cell cluster embedded in extracellular matrix, a large colony, and a naked diploid flagellate. In general, colonies form under nutrient-replete conditions in sufficient light (Brussaard et al.[24]) and enclose photosynthetic non-flagellated cells. Haploid flagellates are associated with colony senescence and decline, are probably involved in sexual reproduction, and represent the life stage that persists through nutrient-deplete conditions. Other *Phaeocystis* species have been only found as solitary flagellates (*P. cordata*, *P. rex*, and *P. scrobiculata*; reviewed in Andersen et al.[46]). Ploidy and typical morphotype sizes are indicated. **d** Estimates of lineage divergence times (node bars: 95% HPD) based on the concatenated phylogeny of 17 proteins (10,766 positions). Note the monophyly of polar strains in blue, coinciding with the last Antarctic reglaciation 12 Mya. **e** Pairwise sequence identity of best blast hits for protein models from *Phaeocystis* and other algal groups prevalent in the marine environment. Compared to diatoms (Pt × Tp, *Phaeodactylum tricornutum* vs. *Thalassiosira pseudonana*), chlorophytes (Cr × Ol, *Chlamydomonas reinhardtii* vs. *Ostreococcus lucimarinus*), and pelagophytes (Aa × Ps, *Aureococcus anophagefferens* vs. pelagophyte CCMP2097), *Phaeocystis* are a recently divergent group (Pa, Pc, Pg, *P. antarctica*, *P. cordata*, *P. globosa*, respectively). **f** Protein orthologous group overlap between *Phaeocystis* reference genomes and other algal groups. Source data are provided as a Source Data file.

worldwide blooms of polar (*P. antarctica*, *P. pouchetii*) and temperate (*P. globosa*) colony-forming *Phaeocystis* are second only to diatom blooms[15,16]. While their blooms can be detrimental to fisheries, aquaculture, and tourism[17], they also play key roles in biogenic fluxes, including substantial vertical transport of carbon from the euphotic zone[18]. Additionally, most species form ecologically important interactions with Acantharia and dinoflagellates[19,20].

*Phaeocystis* employ strategies to cope with biotic and abiotic stress, such as nutrient limitation, reduced illumination, and ocean acidification[21–24]. Furthermore, *Phaeocystis* exhibit a polymorphic life history[25] (Fig. 1c), and while mixotrophy (including bacteriovory) has been shown for solitary flagellates[26–28], colonies developing under nutrient-replete conditions benefit from the bacterial communities associated with their matrix through enhanced iron and vitamin B acquisition[29–31] and efficiently deter predation and viral infection[25,32–34]. Though such adaptations appear crucial for *Phaeocystis*[35], their

molecular regulation is less understood, which could be resolved using reference genomics.

While reports on their global biogeography exist[14,36], they are based on amplicons or partially assembled genomes, and do not elaborate on gene-level adaptation. Here, we present genomic data for thirteen strains of five *Phaeocystis* species (*antarctica*, *cordata*, *globosa*, *jahnii*, and *rex*) collected worldwide. By mapping reads from multiple expeditions and controlled experiments, we compare the biogeography and adaptive strategies of Phaeocystales. We find that morphotype transition, known to be important for *Phaeocystis* in response to environmental conditions, has a genomic context. In particular, strong mitochondrial transcription suggests a mixotrophic lifestyle of some strains under specific conditions. Genome comparisons show considerable expansions in protein-coding content, similarly to other haptophytes, with significant enrichment in several rapidly expanding protein domains. Many of these, such as

transporters, xanthorhodopsins, and sulfotransferases, may underlie the ecological success and biogeochemical impact of the group.

## Results And Discussion

### Repeat-rich *Phaeocystis* draft genomes show various contiguity but comparable coding capacities

Genome completeness and taxonomic coverage are important parameters of genomic resources, and we show that the Phaeocystales dataset satisfies both. Thirteen *Phaeocystis* isolates were sequenced, resulting in haploid assemblies ranging from 89.5 to 199.1 Mbp (Supplementary Data 1). Three of the genomes, *P. antarctica* CCMP1374, *P. cordata* CCMP3104, and *P. globosa* Pg-G(A), hereafter referred to as Phaant1, Phacord1, and Phaglo1, assembled into larger contigs with N50 = 1,556,472, 30,700, and 358,336 bp, respectively. These assemblies were annotated[37,38] based on MMETSP[39] transcriptomes for *P. antarctica* and *P. cordata*, as well as transcriptomic data from a wide array of conditions for *P. globosa*, resulting in 37,567, 33,431, and 29,900 non-overlapping gene models (Methods). Phaant1 and Phaglo1 were comparable in size and contiguity to the *Emiliania huxleyi* CCMP1516 assembly (Emihu1[40], 167.9 Mbp, N50 = 404,808 bp; Supplementary Data 1), but less contiguous than a recently published *P. globosa* genome (129.7 Mbp, scaffold N50 = 6.6 Mbp, 32,618 genes)[41]. While more fragmentary, other culture-derived *Phaeocystis* assemblies had similar gene content, as determined by conserved ortholog and *Phaeocystis*-specific gene searches (Supplementary Note 1). Similarly to Emihu1[40], considerable proportions of *Phaeocystis* genomes are repetitive (Supplementary Note 2, Supplementary Data 2). Specifically, repetitive elements make up 35% (55 Mbp) of Phaglo1 and 50% (101 Mbp) of Phaant1, which partially explains the higher genome size of Phaant1, and the fragmentation of assemblies when using short reads only (Methods). The non-autonomous TIR and long-terminal repeat retrotransposon elements of the TRIM/LARD type are the most abundant putative transposable elements (TEs), the latter found in greater abundance in Phaant1; predominant autonomous TEs in Phaant1 and Phaglo1 belong to Copia and LINE retrotransposon families. As Emihu1, Phaglo1 and Phaant1 also contain high proportions of simple sequence repeats. The organellar genomes are highly complete and show an organization typical for haptophytes[42,43] (Supplementary Note 1), although the plastid genome underwent stop codon reassignment (UGA=Trp), a unique feature among algae and algae-derived apicomplexan parasites[44,45] (Supplementary Fig. 1d, e).

Representatives of other Phaeocystales, such as *P. scrobiculata* and undescribed symbiotic species, remain uncultured[46]. To expand our taxonomic sampling for phylogenomics and biogeography, our analyses also include 21 selected Tara Oceans metagenome-assembled genomes (MAGs)[36]. Whereas MAGs constitute only partial genomes (10.2-54.3 Mbp) lacking organellar and rDNA sequences, they exhibited conserved ortholog scores largely comparable to culture-derived assemblies (9.8-56.9%, mean=40.4%, Supplementary Data 1) and sufficient for downstream analyses.

The comprehensiveness of the Phaeocystales dataset allowed us to reconstruct their phylogeny with great resolution. Our phylogenomic trees are consistent with previous works[46,47] (Supplementary Fig. 1), and, moreover, the 240-protein matrix recovered a highly supported, monophyletic relationship between *P. antarctica* and *P.* cf. *pouchetii*, the latter identified among the MAGs based on a predominantly Arctic distribution (see below). A two-point calibrated timetree placed the split between *P. antarctica* and *P.* cf. *pouchetii* to 12.3 ± 1.27 Mya (mean±95% CI), which coincides with the latest glaciation event in Antarctica[48] (Fig. 1d). Our analyses further suggest a sub-species structure among *P. globosa* strains, representing several independent genotypes, a view supported by mitochondrial genome rearrangements (Supplementary Note 1, Supplementary Fig. 1) and single nucleotide variations[41]. Many Phaeocystales MAGs clearly represent overlooked relatives of cultured *Phaeocystis*, with

phylogenetic affiliations to the polar clade, a broader *P. jahnii* clade, and a more distantly branching clade previously coined sister *Phaeocystis*[36]. Remarkably, we have no morphological and little environmental data concerning this lineage.

In summary, although the architecture of these genomes does not substantially depart from other haptophytes, the data greatly improve the genomic resources for the group and highlight the worldwide diversity of Phaeocystales. We examine how this genomic resource facilitates functional analyses of a wider, uncultured diversity of *Phaeocystis* in situ.

### *Phaeocystis* are globally distributed with lineage-specific preferences

Biogeographic studies of eukaryotes traditionally rely on sequencing short regions of universal marker genes via metabarcoding[49–52] that cannot fully resolve phytoplankton diversity[53] or capture physiological responses. Consequently, there has been an unprecedented accumulation of metatranscriptomic (metaT) and metagenomic (metaG) data from various environments that allow for higher taxonomic and functional resolution and in situ physiological responses of whole communities[54–57]. By adapting a pipeline for genome-wide environmental read mapping[58], we describe the global distribution of Phaeocystales drawing on data collected by multiple cruises[52,56,57,59,60] (Supplementary Data 3).

Altogether, $0.96 \times 10^9$ metaG reads mapped to the combined Phaeocystales assemblies, representing 0.9 % of all processed reads (n = $105.7 \times 10^9$) from 103 worldwide stations (Supplementary Note 3). This is in good agreement with previous works, assigning 0.25–3.72 % of global reads, and at least 4.3 % of global biomass, to *Phaeocystis*[5,13,14,56], and correlates with both published 18S-V9 abundances[50,61] (Pearson's $r(40) = 0.60$-$0.82$, $P < 10^{-4}$, details in Supplementary Data 3) and metaT data (Pearson's $r(75) = 0.41$-$0.99$, $P < 10^{-4}$, details in Supplementary Data 3). Most reads mapped to *P. antarctica*, *P. globosa*, and *P. pouchetii* (27.9 %, 17.3 %, and 13.3 % of the total, respectively), but many of the uncultured Phaeocystales MAGs, including *Phaeocystis* sp. 1, the broader *P. jahnii* clade, and the *Phaeocystis* sister clade (PSC), were also notably abundant. The former two (TARA_AOS_82_MAG_00183, polar clade, 5.34 % total; TARA_ARC_108_MAG_00248, in the otherwise temperate/tropical *P. jahnii* clade, 5.59 %) were largely restricted to polar regions, whereas PSC (11.5 %) occurred throughout temperate and tropical regions. Overall, species abundances were unevenly distributed but noteworthy, and reads mapping to most taxa were found throughout all stations. Polar areas were dominated by *P. antarctica* and *P. pouchetii* (mean=23,602 reads per million, RPM), whereas warmer waters were inhabited by less abundant, more diverse Phaeocystales communities (mean=4,241 RPM) (Fig. 2a; Supplementary Fig. 2). Notably, four MAGs not affiliated with the *antarctica/pouchetii* polar clade appear to have a substantial polar presence, suggesting convergent colonization of cold waters (Supplementary Note 3). Among size fractions, most (61%) Phaeocystales reads were recovered from pico-sized (<5 μm) filters. Colony-forming species associated with larger size fractions under specific conditions, generally low silicate (*P. pouchetii*) or high nitrate, suggesting these conditions promote colony-formation. Specifically in the Arctic and Southern Ocean, where blooms are expected, *P. pouchetii* and *P. antarctica* associated with mesoplankton (>200 μm), indicating the colonial morphotype contributes to large-fraction biomass (Fig. 2a). Early-branching lineages, which are not known to form colonies or symbioses, were mostly found in small (<20 μm) size fractions. Small-sized fraction abundances often positively correlated with ammonium, but not with nitrate (Supplementary Fig. 3a, Supplementary Note 3), and also correlated with temperature, e.g., different temperature preferences were found for *P. globosa* genotypes (Supplementary Fig. 3b). While Phaeocystales are widely recognized as ubiquitous nanophytoplankton, our findings reveal overlooked lineages with varying abundances and environmental

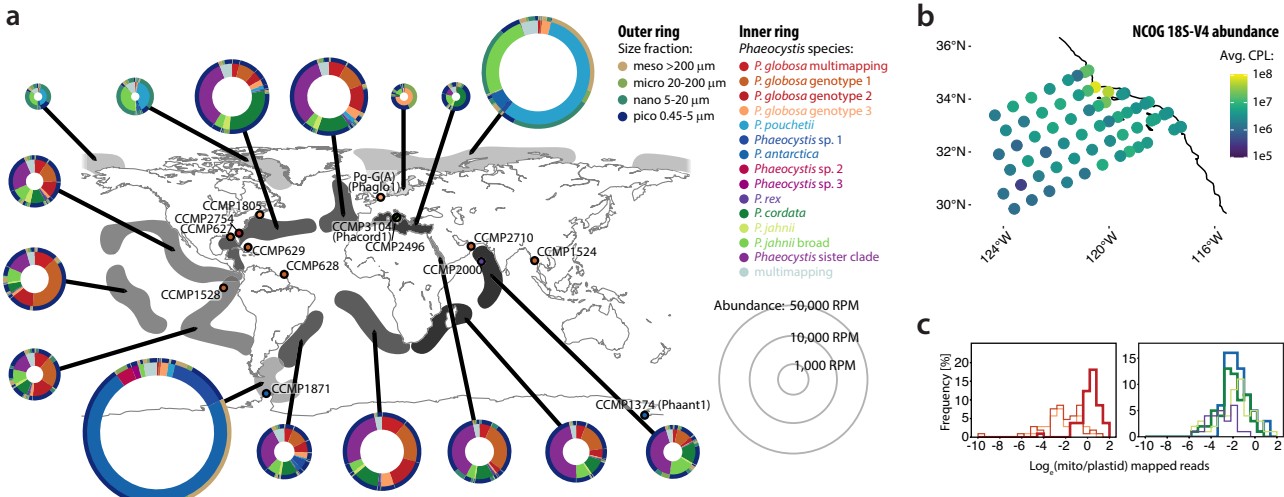

**Fig. 2 | Biogeography of *Phaeocystis* spp. with respect to size fractions. a** The isolation sites of CCMP accessions and their biogeography, based on genome-wide Tara Oceans metagenomic read mapping, normalized to library size, expressed as reads per million (RPM). Data includes numbers for Phaeocystales MAGs[37]. For each oceanic domain, multi-level pie charts are shown, with inner circles representing the contributions of each *Phaeocystis* spp. to the total read abundance, and outer circles representing the proportions of reads from different size fractions mapping to these species, clockwise according to the color legend. In most oceanic domains, *Phaeocystis* spp. occur as flagellates among nano- and picoplankton. **b** Copies per liter (CPL) abundance of 18S-V4 of *Phaeocystis* OTUs in the CalCOFI/NCOG data. **c** Log_e of the ratio of mitochondria- and plastid-mapping reads for *Phaeocystis* spp. where both organellar genomes are available. Left panel plots *globosa* genotypes, right all other species. Thicker lines mark the three species with the highest number of mitochondrial-mapping reads, *P. antarctica*, *P. cordata* and *P. globosa* genotype 2. Note a somewhat bimodal distribution for *P. cordata*, suggesting higher mitochondrial activity is condition specific. Source data are provided as a Source Data file.

specializations, shaped by complex evolutionary histories. Given that *Phaeocystis* is currently treated as a single PFT in global biogeochemical models, such hidden diversity could have important implications. If members of the clade differ in ecological roles and functional traits, model predictions may be affected. Our results emphasize the value of incorporating multiple, data-informed groups into future *Phaeocystis* experimental frameworks – paralleling efforts to refine models through strain-specific thermal niches of *E. huxleyi*[62].

Next, we addressed correlation with environmental variables using CalCOFI (NCOG) metaT data (Supplementary Data 3) that comprises relatively large temporal and biogeochemical variability across 307 samples in the California Current Ecosystem (CCE)[52]. In these data, *P. globosa*, *P. cordata* and PSC were constitutively present without bloom events (Fig. 2b). Hierarchical clustering of *Phaeocystis* transcript orthogroups identified 9 super-clusters, which explained ~92% of their transcriptomic variance. Temperature and depth were the strongest drivers of this variance, and several clusters showed changes in predicted transcriptomic proportion across a range of temperatures (-11–18 °C) (Supplementary Fig. 4). Pfams associated with the super-clusters having relatively increased transcript proportion at higher temperatures corresponded to anabolic and photosynthesis-related functions (Supplementary Data 4), whereas super-clusters with relatively decreased transcript proportion contained few exclusive biological functions. The most remarkable of these is MPV17, a mitochondrial DNA copy number and maintenance protein, suggesting a switch from mitochondrial to plastid-driven metabolism over this temperature (and depth) gradient (Supplementary Fig. 4). Supporting this notion, a similar analysis of euKaryotic Orthologous Groups (KOG) terms clearly identified decreased (mitochondrial) energy metabolism-related transcription with increasing temperature, and a concomitant increase in transcripts involved in translational and post-translational processes (Supplementary Fig. 4).

Notably, while metaG reads sparsely mapped to mitochondrial genomes, with about 1-3 mitogenome copies per haploid genome (Supplementary Fig. 5b), we found mitochondrial metaT reads in most stations (median=11.5 RPM) (Supplementary Fig. 2). Most (95.1 %) of these reads, largely from smaller size fractions, mapped to only three genomes, Phaant1, *P. globosa* genotype 2 and Phacord1, which also exhibited much higher mitochondrial-to-plastid read ratios (Fig. 2c). In these strains, mitochondrial transcription clearly has an important function, perhaps supporting flagellar or haptonemal motility, and responds to environmental cues, such as iron and nitrogen availability, particularly at lower occurrences (Supplementary Fig. 5a). Additionally, signatures of heterotrophy vary for Phaeocystales unigenes detectable across Tara Oceans stations, suggesting metabolic flexibility (Supplementary Fig. 5c, d). Motile cells might facilitate an ecological advantage to *Phaeocystis* via mixotrophy, i.e., supplementing nutritional requirements with compounds from prey or organic matter, especially when in competition with diatoms and dinoflagellates, which also employ various strategies to obtain nitrogen[63,64]. Consistent with bacteriovory[28], transcripts associated with lysosomes and membrane trafficking are significantly increased in *P. globosa* at stations with high mitochondrial-to-plastid transcription (Supplementary Data 5). According to metabolic models, *P. antarctica*, *P. globosa*, and *P. cordata* each support mixotrophic growth, although respond differently to various forms of nitrogen, perhaps priming them for different nutrient acquisition mechanisms (Supplementary Note 4, Supplementary Fig. 6). We hypothesize that variable rates of mixotrophy and mitochondrial transcription contribute to this flexibility and affect ecological niche partitioning between *Phaeocystis* lineages.

Comparisons of Pfam expression profiles in temperate (CCE) and polar (Arctic, Southern Ocean) biotopes additionally show that iron and B12 shortage strongly shape the physiology of local Phaeocystales communities (Fig. 3e–i, Supplementary Note 5, Supplementary Data 6). Among the hundreds of differentially abundant Pfams, various iron-responsive domains are particularly highly expressed in the Southern Ocean (Supplementary Note 5), although the overrepresentation and widespread expression of iron-responsive proteins[27] (ISIPs,

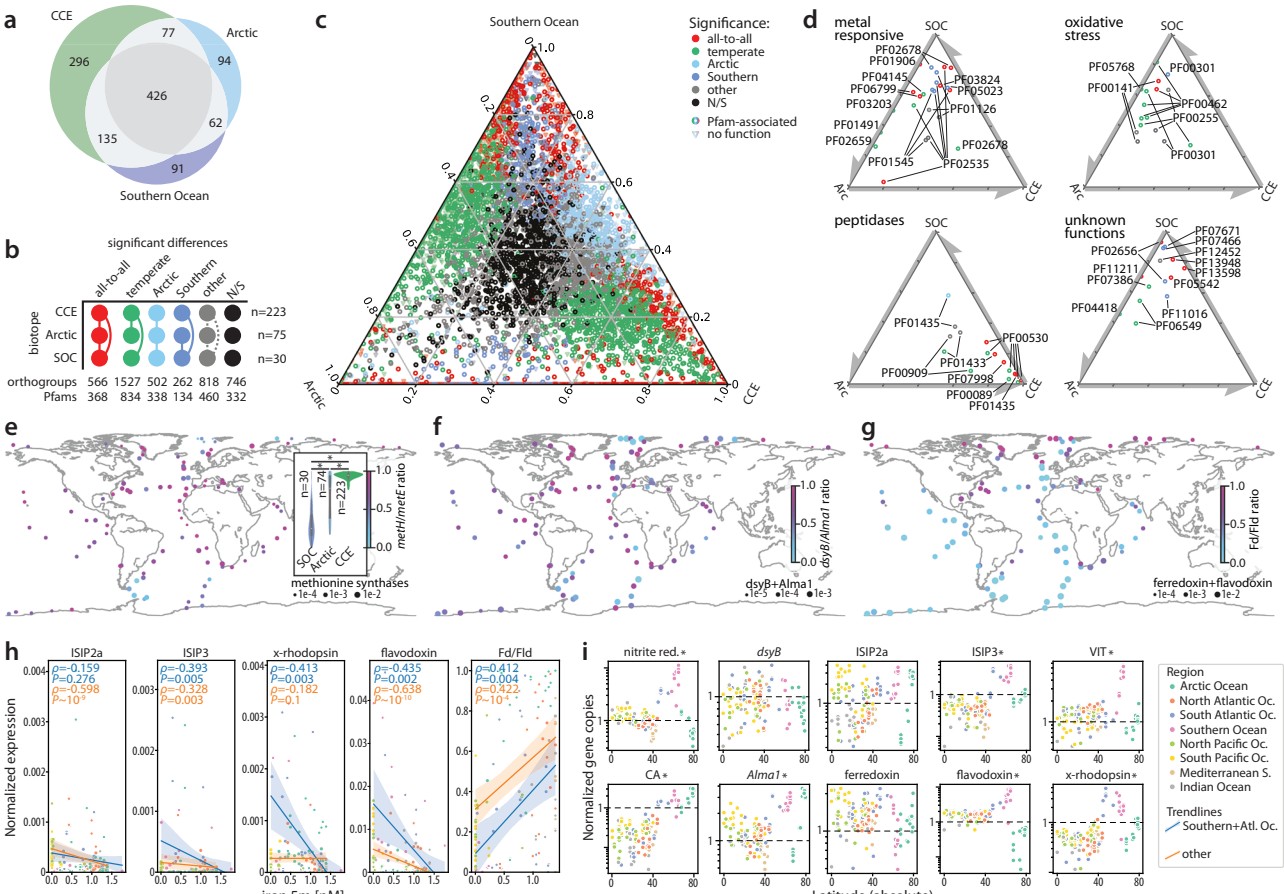

**Fig. 3 | Phaeocystales functional profiles in temperate and polar biotopes.**
**a** Venn diagram of Pfams for the top 1000 orthogroups with highest average expression (TPM) in each of the three analyzed biotopes (CCE, California Current Ecosystem; Arctic; SOC, Southern Ocean). **b** Patterns of significantly different enrichment among the three biotopes/datasets (nodes). Edges represent significant difference; colors correspond to colors in panels c-d. Numbers represent counts of annotated orthogroups and Pfams with respective significance patterns ($p$-adjusted Mann-Whitney test; Methods). **c** Relative normalized expression (TPM) of 7,316 orthogroups in three biotopes, colored by significance of expression difference. Circles and triangles mark orthogroups with and without Pfam annotations, respectively. **d** Relative normalized enrichment of selected Pfams in three biotopes, colored by significance of enrichment difference as in **c**. **e** Relative expression of B$_{12}$-dependent (*metH*) and B$_{12}$-independent (*metE*) methionine synthases ($metH/(metH+metE)$) globally and in the three biotopes from panels **a–d**. * – significant difference between biotopes ($p$-adjusted Mann-Whitney test; box-and-whisker plots within the violin plots show median, interquartile range, and 1.5*IQR

values; Supplementary Note 5). **f** Relative expression of organosulfur biosynthesis enzymes methylthiohydroxybutyrate methyltransferase (*dsyB*) and DMSP lyase (*Alma1*) showing a relative enrichment in *Alma1* expression near a Svalbard bloom. **g** Relative expression of iron-indicator markers ferredoxin (Fd, PF00111) and flavodoxin (Fld, PF00258). Flavodoxin, expressed in iron-limiting conditions, is widely utilized by *Phaeocystis*. **h** Expression of iron-responsive proteins, colored by oceanic region, and their trend lines in Southern+Atlantic Ocean and other oceanic domains (see legend in panel **i**). Values normalized to total Phaeocystales expression; error bands represent 95% CI to the corresponding linear regressions. Two-sided Spearman's *rho* and *p*-values are shown in the upper left corner for the Southern+Atlantic Ocean (in blue, n = 49) and the other data (in orange, n = 82). The expression of all genes in panels **e–h** was normalized to Phaeocystales total. **i** Gene expansion of selected gene families as a function of latitude (* – genes with adjusted *p*-value < 0.001; Supplementary Note 5). Gene copies normalized to length and single-copy gene loci. CA, carbonic anhydrase; Nitrite red., nitrite-sulfite reductase; VIT, vacuolar ion transporters. Source data are provided as a Source Data file.

xanthorhodopsin, flavodoxin) suggests that iron-saving adaptations are widely employed by Phaeocystales (Fig. 3g–i). The Southern Ocean is also unique for the high expression of B$_{12}$-independent methionine synthase[65], which enables *Phaeocystis* to circumvent the shortage of this essential vitamin (Fig. 3e). The interactions between *Phaeocystis* and other phytoplankton, particularly the contribution of mixotrophy to their macro- and micronutrient budget, are therefore important factors of succession during blooms in different regions and warrant future investigation.

### *Phaeocystis* spp. encode distinctive profiles of rapidly evolving Pfam families

The genome annotations of Phaant1, Phacord1, and Phaglo1 allow a comprehensive quantification of gene families. In functional terms, 45.5 %, 61.2 %, and 49.1 % of Phaant1, Phacord1, and Phaglo1 genes, respectively, could be assigned a Pfam annotation. Approximately

one-third of the annotations involved post-translational modification, signal transduction, and intracellular trafficking (Fig. 4a). Consistently, transcripts associated with these processes recruited most environmental reads (Supplementary Fig. 7). Among the most abundantly mapped were also genes with functions related to cytoskeleton, photosynthesis, and translation. Transporters represent ~3.6–4.4 % genes in *Phaeocystis*, with some families particularly numerous among haptophytes (ABC, DMT, MFS; Supplementary Note 4; Supplementary Data 7).

To explore the evolutionary origin and dynamics of their genomes, we performed ortholog clustering with representative databases of eukaryotic, prokaryotic, and viral sequences (Methods). Whereas *P. antarctica* has been shown to encode ~36 % accessory orthogroups[26], our analyses additionally suggest *Phaeocystis* possess 25–40 % accessory orthologous groups (OGs) missing in other algae (e.g., other haptophytes, stramenopiles, and dinoflagellates) (Fig. 1). The accessory OGs encompassed multiple regulatory Pfams (zinc

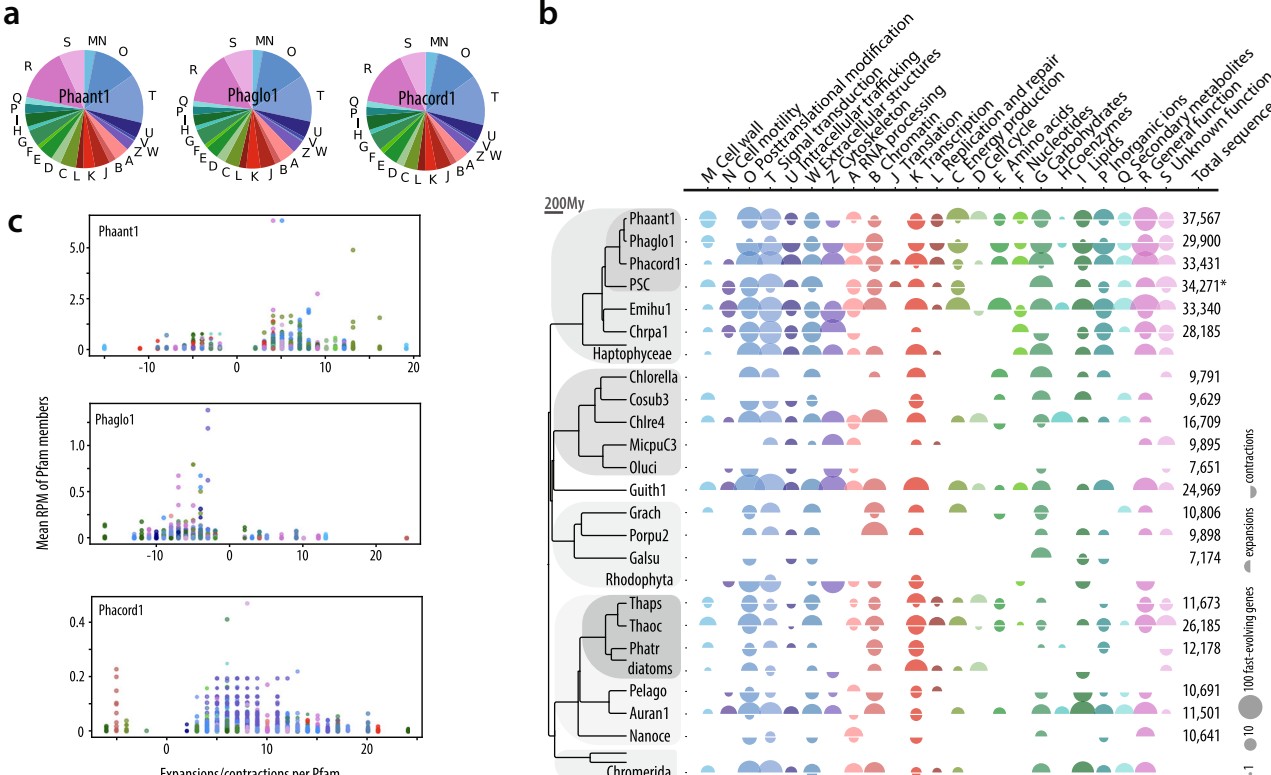

**Fig. 4 | Functional analysis of fast-evolving Pfam families in *Phaeocystis* spp. and other algal genomes. a** Global fractions of KOG biological functions in *Phaeocystis*. Functional groups (letters) correspond to panel **b**. **b** For each phylogenetic node, upper and lower semi-circles represent, respectively, the number of gained and lost genes belonging to fast-evolving Pfam families, classified by KOG biological functions. The total number of genes in analyzed algal genomes is shown for comparison, clearly distinguishing haptophytes *Emiliania huxleyi* and

*Phaeocystis* spp. from other algae in terms of gene richness. Only non-overlapping gene models were used for *Phaeocystis* (i.e., not isoforms). Based on a multigenic analysis, the ultrametric timetree on the left indicates the approximate divergence times of shown species in millions of years (My). **c** Average abundance of genes (RPM, reads per million) belonging to fast-evolving Pfams from Tara Oceans metaT data, plotted against the number of gained/lost genes per respective family. Source data are provided as a Source Data file.

fingers, Myb-like, EF-hand, and protein kinases) but no biological or molecular functions were significantly enriched, leaving the overall importance of the accessory portions of *Phaeocystis* genomes largely unknown. Phylogenetic profiling of all OGs revealed 183 horizontal gene transfer (HGT) events (totaling 512 genes in the three reference genomes, Supplementary Data 8), i.e., cases where *Phaeocystis* genes were robustly nested within clades of non-haptophyte origin (Fig. 5a, f). Most HGTs originated in stramenopiles, dinoflagellates, and opisthokonts, and functionally contribute to a variety of functions (Fig. 5, Supplementary Note 6). This illustrates the substantial, likely stochastic, gene flow between marine biota and Phaeocystales, corresponding to their cosmopolitan distribution.

Next, we compared Pfam enrichment between main algal lineages and found divergent patterns. For instance, diatoms exhibit rapid evolution, both before and after their radiation (Supplementary Data 8), primarily in transcription-related domains (e.g., helicase, high mobility group, and heat-shock factor) (Fig. 4b). This is consistent with diatoms' reliance on dynamic transcriptional and post-transcriptional regulation of gene expression[55,66]. Other algal groups showed only minor or species-specific expansions in transcription-related families (stramenopiles other than diatoms, chlorophytes), or expansions in post-translational modification and signal transduction (haptophytes, chromerids, *Guillardia*), suggesting regulation on translational and post-translational level could be more substantial here (Fig. 4b).

In haptophytes, significantly expanded Pfams belong to most major biological processes, hinting that they rely on gene duplication. Haptophyte genomes indeed encode 2–3× more genes than smaller-genome diatoms (Fig. 4b). The highest Pfam enrichment was

found in Emihu1, Phacord1, and PSC, which are among the most gene-rich genomes in our comparison (Fig. 4b). Importantly, *Phaeocystis* spp. showed significant Pfam expansions that might underlie their specific biology. While gene copy numbers need not correlate with enhanced functionality, gene family expansions in inflated genomes often lead to elevated expression or functional novelty (e.g.[67,68]). One group of expanded families consisting of glycoside transferases, sugar transporters, fibronectins, sulfotransferases, and exostosins, probably underlies the formation of extracellular structures (such as scales and star-shaped filaments[25]). Specific expansions were also seen in photosynthesis (e.g., xanthorhodopsins, redoxins), compound transport, and protein modification/signal transduction (Supplementary Note 6, Supplementary Data 8), the latter potentially having a major role in regulation[55]. Phaglo1 showed lower domain richness than other *Phaeocystis* or PSC (Fig. 4b, Supplementary Data 8), with significantly enriched Pfams having putative extracellular functions (e.g., von Willebrand, carbohydrate sulfotransferases, and C-lectin). While they are not exclusive to colony-forming *Phaeocystis*, von Willebrand proteins were found to be iron-responsive and hypothesized to participate in colonial matrix formation[30]. Phaant1 showed expansions in most functional classes, and environmental data suggest they are expressed in situ (Fig. 4c, Supplementary Data 8). As such, they likely represent adaptive portions of the genome and contribute to the ecological success of *P. antarctica* in the Southern Ocean. Notably, nitrite/sulfite reductase and carbonic anhydrase domains are significantly expanded in Phaant1 and other Southern Ocean Phaeocystales (Fig. 3i, Supplementary Note 5), perhaps enhancing the assimilation capabilities of inorganic nitrogen, sulfur, and carbon. Furthermore, vacuolar ion

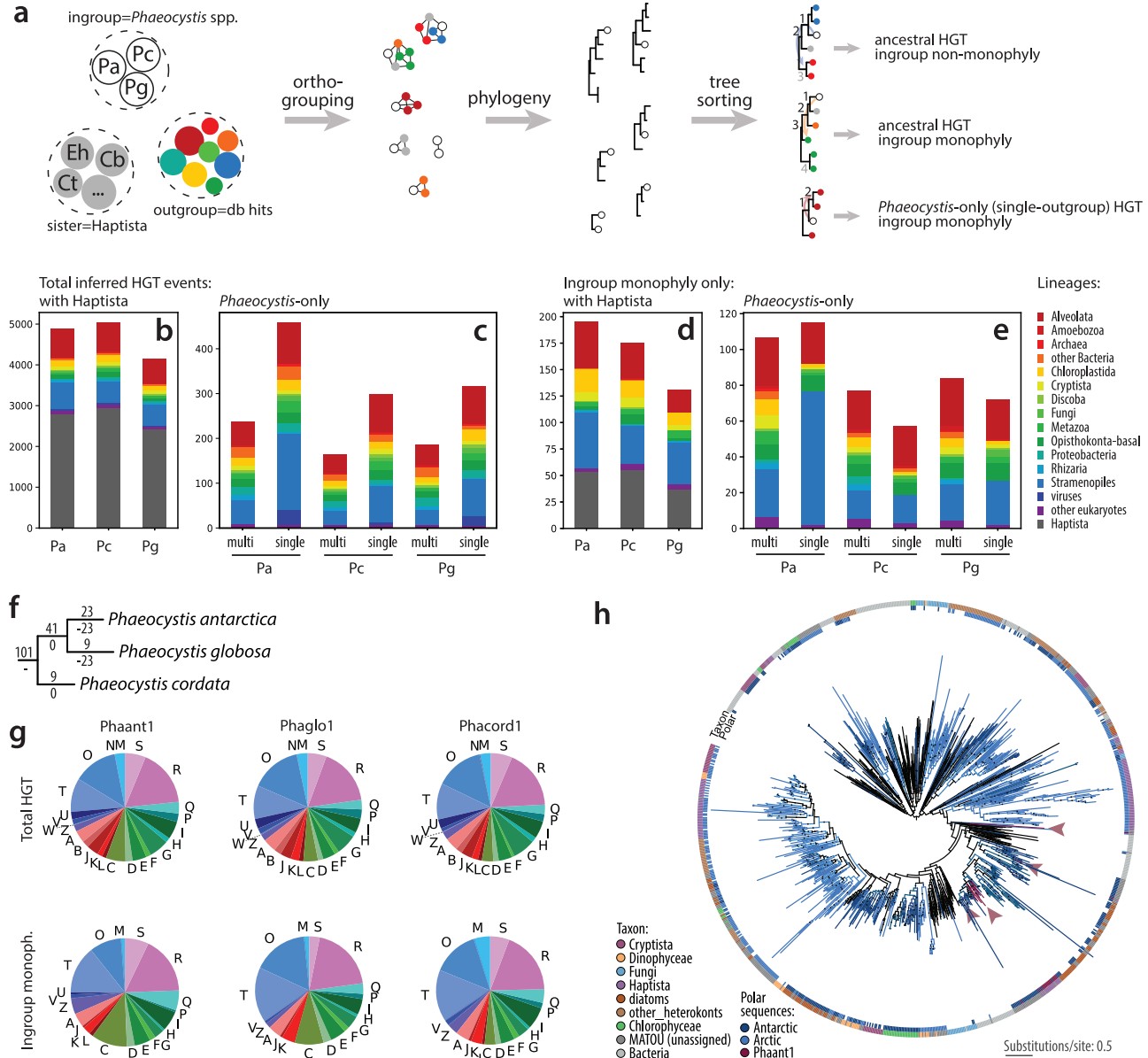

**Fig. 5 | Horizontal gene transfer (HGT) events in *Phaeocystis* draft genomes. a** The workflow overview (Methods). **b–e**, The left and right panels summarize the taxonomic composition for all detected HGT events (referred to as total, i.e., including cases where Haptophyta/Phaeocystales possibly donated genes) and those filtered by ingroup monophyly, respectively. Pa - *P. antarctica*, Pg - *P. globosa*, Pc - *P. cordata*. Note that the vast majority (98.2%) of Alveolata sequences detected in the HGT clades belonged to Dinoflagellata. Hundreds of additional candidate HGTs, including viral sequences, were found, though the direction of gene transfer to or from *Phaeocystis* could not be confidently inferred. **b, d**, Numbers of genes, where haptophyte sequences were present in the HGT clade, and these events likely pre-date the split of Phaeocystales from other Prymnesiophyceae. **c, e** Numbers of

HGT genes exclusive to Phaeocystales. Stacked bars show the contributions of various lineages to these HGT events. **f** The number of HGT-originated orthologous groups gained and lost at each lineage of *Phaeocystis*. **g** Functional annotations of the total HGT pool. **h** Phylogeny of ice-binding proteins. Each tip represents a sequence with a taxonomic affiliation according to the color legend, or an unassigned sequence from the Tara MATOU v1 database (dark grey). Sequences with predominantly polar occurrence or found in polar algae are marked in the inner band; clades found in *P. antarctica* are marked by purple arrowheads. Dataset modified from[73], redundant sequences removed. Source data are provided as a Source Data file.

transporters (VIT/Ccc1) could participate in $Fe^{2+}$ uptake and storage[69]. The over-representation of additional iron-responsive and organo-sulfur metabolism genes compared to warmer latitudes (Fig. 3i, Supplementary Note 5) suggests that Southern Ocean-specific expansions may underlie the observed higher expression levels, likely an adaptation to chronic nutrient depletion[70,71]. Noteworthy adaptive novelties of *P. antarctica* include ice-binding proteins (IBPs, Fig. 5), specifically expanded in polar algae[72,73]; the horizontal transfer and expansion of IBPs were also likely crucial for Southern Ocean colonization.

## Viral footprints are found integrated in *Phaeocystis* genomes

*Phaeocystis* are known to host several nucleo-cytoplasmic large DNA viruses (NCLDVs, e.g., Phaeocystis globosa virus, PgV[74,75]), including Mesomimiviridae[76], which in turn are host to virophage-like elements as Polinton-like viruses (PLVs[77-79], e.g., Gezel-14T[79]). Some viruses integrate into eukaryotic genomes, revealing ancient or cryptic viral-host interactions[80,81]. Meanwhile, integrated (endogenous) virophages were shown to protect eukaryotic host populations by inhibiting the replication of their NCLDV host[82-84].

We found several loci in Phaant1 and Phaglo1 (but not Phacord1) to contain multiple hallmark genes of PLVs/virophages. Sequence comparison and phylogeny suggest they represent two groups that most resemble PLVs, hereafter named Phaeocystis endogenous PLV (PePLV) (Supplementary Fig. 8a, b). PePLV2 copies are Phaant1-specific and heavily truncated, whereas PePLV1 copies are apparently complete insertions (7 in Phaglo1, 2 in Phaant1, ranging 20–27 kbp; Supplementary Fig. 8e), with more conserved genes and terminal inverted repeats. Corroborating recent insertions, these copies are inserted in different genomic contexts and are subject to frequent recombination (Supplementary Fig. 8d, e; Supplementary Note 2). MetaT read recruitment to PePLV loci correlated with peak *Phaeocystis* abundance, suggesting that these virophage-like loci respond to infection by certain NCLDVs (Supplementary Fig. 8f, g). Similarly, virophage promoters in *Cafeteria roenbergensis* are type-specific and only respond to certain Cafeteria viruses[85]. Interestingly, meiotic genes are also expressed at stations with PLV/NCLDV-related expression (Supplementary Fig. 8g). Additional clades of PLVs/virophage sequences were found in Phaeocystales genomes, though only *P. antarctica* and *P. globosa* retain full-length copies (those of PePLV1). Given their phylogenetic relationship with other haptophyte PLVs/virophages, PePLVs apparently co-evolve with *Phaeocystis* (Supplementary Fig. 8b).

We also found endogenous NCLDV; Phaant1 and Phaglo1 endogenous NCLDVs (PaeNCLDV and PgeNCLDV, respectively) are highly colinear and span 47.4–53.6 kbp (Supplementary Fig. 8a). Among the predicted ORFs, only four of six core NCLDV proteins were identified (Supplementary Note 2; Supplementary Data 9). Nevertheless, their phylogeny suggests a close relationship with Yaravirus-like viruses[86] (Supplementary Fig. 8b), which also have relatively small genomes and lack some core NCLDV proteins. These genomic footprints show that *Phaeocystis* are host to both Mimiviridae-related and Yaravirus-like NCLDVs.

To better understand the interactions of NCLDV with its *Phaeocystis* host, we compared transcriptomic data from time points tracking the infection by exogenous PgV. PgV-07T[75] infecting *P. globosa* Pg-(G)-A triggered a distinctive response at 4 and 8 hours post-infection (hpi), whereby relatively few metabolic pathways were affected, while ribosomal proteins were consistently and significantly increased (Supplementary Note 7, Supplementary Data 10). By 24 hpi, host biological processes halted, consistent with PgV's infection cycle ~30 hours[75]. PePLV loci seemed unaffected by PgV-07T infection, suggesting type incompatibility with the PgV strain used. PgV-07T nevertheless elicits similar responses as the rather distantly related EhV infecting *E. huxleyi*[87–89], with similarities clearly stemming from analogous requirements for virion production.

### Nitrate supplementation and dark-light transition induce strong transcriptomic reallocation

To improve gene model prediction for Phaglo1 and establish preliminary expression profiles with higher sensitivity than in environmental samples, we additionally produced transcriptomic data across several growth conditions pertaining to colony development and nutrient and light availability (Supplementary Note 7). Colony development appears to be supported by metabolic rearrangements towards photosynthesis and exopolymer biosynthesis (Supplementary Data 10). Nitrate supplementation (880 $\mu$M $NO_3^-$ versus 0.37 $\mu$M $NO_3^-$ ambient concentration) triggered anabolic responses including shifts in nitrogen compound transporters that were partially mirrored by energy-saving mechanisms in cells entering a stationary phase (Supplementary Data 10), while ammonia amendment (100 $\mu$M $NH_4^+$) resulted in negligible changes. Major changes were also observed in response to light after prolonged (67 hours) darkness, largely involving photosynthetic pathways, protein expression and trafficking, and a transition affecting flagellar motility (Supplementary Data 10).

Contrary to previous studies, which found haptophytes generally not capitalizing on rapid nutrient inputs[55], we find that light-transition and nitrate supplementation elicit dynamic transcriptional responses in *P. globosa* (Supplementary Fig. 7). Genomic, as well as culture-based and environmental transcriptomic evidence showcase the usefulness of this genomic resource, and highlight that Phaeocystales actively employ diverse, functionally overlapping molecular tools to cope with nutrient limitation and biological stressors. The expression levels of these genes, or their ratios, such as *metH/metE*, ferredoxin/flavodoxin (Fd/Fld), and *dsyB/Alma1*, could serve as biomarkers for assessing the physiological state of *Phaeocystis* communities.

In summary, by integrating genomic data with environmental information, genome-assisted biogeography provides a more detailed understanding of the factors driving species distribution across space and time[36,58,90,91]. We uncover that Phaeocystales are both more diversified and abundant than previously thought, employing cosmopolitan or cold-water specialist strategies. Their nuanced life histories, likely involving mixotrophy, impart an advantage over competitor phytoplankton or their predators and pathogens. Furthermore, *Phaeocystis* genome evolution is accompanied by substantial gene family expansions, possibly underlying additional fundamental but elusive biological processes (e.g., Fig. 3d, ref. 92). Their functional range extends beyond the confines of individual cells and is modulated by external cues, which highlights the remarkable adaptability of *Phaeocystis*, ultimately contributing to their ecological success. A deeper knowledge of these responses is key to our understanding of their true role in the ever-changing ocean.

## Methods
### Cultivation
*Phaeocystis* strains from the National Center for Marine Algae and Microbiota (CCMP) and the Royal Netherlands Institute for Sea Research (NIOZ) were cultivated in natural seawater with L1 supplements in 14 h:10 h (light:dark) diel cycles at 16 °C (*P. globosa* CCMP1805, NIOZ Pg-G(A)), 20 °C (*P. cordata* CCMP3104, *P. globosa* CCMP628, −629, −1524, −1528, −2754, *P. jahnii* CCMP2496) or 24 °C (*P. globosa* CCMP627, −2710, *P. rex* CCMP2000) and inoculated bi-weekly. The media and light regime for *P. antarctica* strain CCMP1374 were the same, but the culture was grown at 4 °C and inoculated every three weeks. For PgV-07T infection experiments, *P. globosa* strain Pg-G(A) was cultivated in Mix-TX medium in 16 h:8 h diel cycles at 15 °C.

### Genome sequencing and assembly
Genomic DNA isolation and library preparation: 1) *P. antarctica* strain CCMP1374: refer to SAMN00120141; 2) *P. globosa* strain Pg-G(A): refer to SAMN10985124; or 3) other strains: CTAB extraction; 100 ng of DNA was sheared to 803 bp using the Covaris LE220 (Covaris) and size selected using SPRI beads (Beckman Coulter). The fragments were treated with end-repair, A-tailing, and ligation of Illumina compatible adapters (Integrated DNA Technologies) using the KAPA-Illumina library creation kit (KAPA biosystems). qPCR was used to determine the concentration of the libraries, which were then sequenced on an Illumina HiSeq 2500 [CCMP1374, Pg-G(A)] or NovaSeq 6000 (other strains) platform. The prepared libraries were quantified using KAPA Biosystems' next-generation sequencing library qPCR kit and run on a Roche LightCycler 480 real-time PCR instrument.

*P. globosa* strain Pg-G(A) assembly v2 (Released 12/2014): Main assembly was performed using ARACHNE[93] with 30.32× MiSeq data and 24.1× Sanger sequence. This release also used ~18× of PACBIO reads for gap patching. Gaps were patched by first breaking the assembly into contigs >1 kbp. 1kbp of sequence was trimmed off contig ends and the trimmed portion was broken into 100mers. The 100mers were aligned to the PACBIO reads using the short-read aligner BWA v0.7.8[94], and individual PACBIO reads were mapped to specific contigs. PACBIO reads spanning a gap (consecutive >1kbp contigs) were

aligned to the gap and gaps having more than 5 PACBIO reads aligned to them were patched. Patching consisted of assembling the reads crossing a gap using QUIVER and the assembled sequence was patched in. A total of 7,019 gaps were patched, with a total of 2,622,918 bases added to the assembly.

Misassemblies were also assessed using the PACBIO reads by looking for PACBIO reads where >1kb regions of the read aligned to different scaffolds. A total of 24 misjoins were identified and the breaks made. The reads used to make the breaks were then used to make the joins. Only joins that had enough reads supporting them were joined. A total of 12 additional joins were made using the PACBIO reads. Additionally, homozygous SNPs and INDELs were corrected in the release sequence using ~27× of Illumina reads (2×250, 800 bp insert).

*P. antarctica* strain CCMP1374 assembly v2 (Released 5/2017): Main assembly performed using MECAT v1.0[95] and the resulting sequence was polished using ARROW. A 4Kb LFPE paired-end library was aligned to the assembly and fragment coverage at each base was computed (average clone coverage was ~1500×). A drop in fragment coverage below 20× indicated a misjoin in the MECAT assembly. A total of 60 breaks were made on the MECAT assembly. Homozygous SNPs and INDELs were corrected in the release sequence using ~80× of Illumina reads (2×250, 800 bp insert).

Other strains assembly: Libraries were sequenced on an Illumina NovaSeq 6000 sequencer using NovaSeq XP V1 reagent kits, S4 flow-cell, following a 2×151 indexed run recipe. The obtained reads were corrected and normalized to a sequencing depth of 80 using bbnorm of the package BBTools v38.63 (sourceforge.net/projects/bbmap). Preliminary assemblies were created by SPAdes v3.11.1[96] with default settings. Then, bacterial contaminants were identified by DIAMOND blastx v0.9.30.131[97] against the NCBI-nr database and a reference haptophyte database (genome.jgi.doe.gov); contaminant reads were removed by bbmap and the genomes reassembled by SPAdes with increased k-mer length (-k 21,33,55,77,99,111). Plastid and mitochondrial contigs were identified by sequence homology searches and removed; they were assembled separately using iterated read mapping/SPAdes assembly, followed by manual curation. The chromosomal assemblies were decontaminated to remove: a) any sequences with >50% query coverage and >70% percent identity to bacterial accessions in GenBank-nt; b) any contigs having read coverage <4× or >400×, i.e., ~10× less or more than the average coverage for SPAdes assemblies (Supplementary Data 1). Genome completeness was assessed using CEGMA v2.5[98] and BUSCO v5.7.0 in genome mode with an Augustus (Web Server, accessed 18 June 2022) prediction model trained with Phaant1 gene models and the eukaryota_odb10 (v2024-01-08) set of conserved orthologs[99,100]. Genome size estimates from raw reads were calculated using GenomeScope 2.0[101] with k-mer sizes 21, 23, and 25.

The assemblies were deposited at PhycoCosm[38], in DDBJ/ENA/GenBank, and OSF (Data Availability).

### Genome annotation
Nuclear gene models were predicted by two different JGI annotation pipelines using a similar combination of ab initio, protein homology-based, and transcriptome-based algorithms (Supplementary Data 1).

**Phaglo1** and **Phaant1** followed the JGI Plant annotation pipeline (IGC). Transcript assemblies were made from Illumina RNA-seq reads using PERTRAN (Shu, Goodstein, and Rokhsar; unpublished), which conducts genome-guided transcriptome short read assembly via GSNAP v2019-09-12[102] and builds splice alignment graphs after alignment validation, realignment, and correction. Subsequently, PASA v2.0.2[103] was used to align transcript assemblies. A repeat library was created from de novo repeats predicted by RepeatModeler v2.0.4[104]. The predicted repeats underwent functional analysis through Inter-ProScan v5.39-77.0[105], incorporating the Pfam[106] and PANTHER[107] databases. Any repeats that displayed significant hits to protein-coding

domains were subsequently excluded from the repeat library. Finally, the constructed species-specific repeat library was used to soft-mask the genome with RepeatMasker v4.1.2 (Smit et al.; http://www.repeatmasker.org). Putative gene loci were determined by transcript assembly alignments and/or EXONERATE v2.4.0[108] alignments of proteins from genomes available on PhycoCosm[38] (v2.6) (algae *Bigelowiella natans* v1.0, *Emiliania huxleyi* v1.0, *Thalassiosira pseudonana* v3.0, *Phaeodactylum tricornutum* v2.0, *Ectocarpus siliculosus* v1.0, and oomycete *Phytophthora ramorum* v1.1) and Swiss-Prot release 2015_11 of eukaryote proteomes to repeat-soft-masked genomes, with up to 2 kbp extension on both ends unless extending into another locus on the same strand. Gene models in each locus were predicted by homology-based predictors, FGENESH+ v3.1.1[109], FGENESH_EST v3.1.1 (similar to FGENESH + , but using EST to compute splice site and intron input instead of protein/translated ORF), EXONERATE v2.4.0, PASA v2.0.2 assembly ORFs, and AUGUSTUS v3.3.3[110] trained on the high confidence PASA assembly ORFs and with intron hints from RNA-seq read alignments. The best-scored predictions for each locus were selected using a composite homology score Cscore (a protein BLASTP[111] score ratio to the mutual best hit BLASTP score and protein coverage is the percentage of protein aligned to the best of homologs) and protein coverage (the percentage of protein aligned to the best of homologs). The selected gene predictions were improved by PASA by adding UTRs, splicing correction, and alternative transcripts. PASA-improved transcripts were selected if their Cscore and protein homology coverage were >= 0.5, or if covered by RNA-seq. For gene models whose CDS were overlapped by repeats by more than 20%, their Cscore had to be at least 0.9 and homology coverage at least 70% to be selected. Gene models without strong transcriptome and homology support, and with proteins > 30% overlapped by transposon-specific Pfam domains, were removed. Incomplete gene models, low homology supported without fully transcriptome-supported gene models, short single exon (<300 bp CDS) without protein domains nor good expression, and repetitive gene models without strong homology support were manually filtered out. Primary transcripts and alternative isoforms (secondary transcripts) from selected final PASA improved loci were imported to PhycoCosm, with PrimaryTranscripts (longest-at-locus) forming the GeneCatalog available for genome analysis and potential manual curation.

**Phacord1** gene models were produced using the JGI Fungal/Algal annotation pipeline[37,112,113] modified for lack of associated transcriptomic data, similar in approach to IGC. Repeats were masked using a combined a de novo RepeatScout v1.0.5[114] and a standard RepBase v25.03 libraries of algal and plant repetitive elements. Proteins from public databases (NR, Swiss-Prot) and related species (Phaglo1 and Phaant1) were mapped onto the masked Phacord1 genome assembly using BLASTx[111] with e-value < 1e-5. These alignments served as seeds for homology-based gene prediction. Transcriptome assemblies from accession MMETSP1465 [*Phaeocystis cordata* RCC1383][115] were aligned with BLAT v35[116]. Gene models were predicted using a combination of ab initio, protein homology-based, and transcriptome-based algorithms (FGENESH v3.1.1[109], GeneMark-ES v2.1[117], GeneWise v4.0[118], combEST v2015[119]; Supplementary Data 1) and improved with estExt (Grigoriev, property of the Lawrence Berkely National Laboratory, not publicly available) using RNA contigs alignment, adding additional CDS exons and untranslated regions (UTRs), and correcting gene structures that disagreed with aligned transcript splicing. Gene models that are similar to transposable element (TE) proteins, have TE PFAM domain families, or lie within repeat-masked regions have been removed. To select the best representative gene model, at loci where multiple gene predictors produced overlapping models, we employed a heuristic approach based on a combination of protein homology and transcriptome support[113]. Specifically, homology support was measured by alignments with the best BLASTp hit

from NR, Swiss-Prot, or PhycoCosm, where only alignments with BLASTp score > 50 and that covered at least 25% of length of gene models were considered. Transcriptome support was measured by correlation coefficient (CC) of the predicted gene model relative to mapped transcripts overlapping with the models the average of all CCs computed for each overlapping transcript. Each gene model was assigned the following empirical score: S = Sblast * (cov1 * cov2 + CCa), where Sblast was the combined BLASTp score of alignments between the gene model and its protein homolog, cov1 and cov2 were alignment coverages for the model and homolog respectively (0 <= cov1, cov2 <= 1), and CCa was the average CC between the model and overlapping transcripts. At each locus, a model with the highest score was selected, and all other models, including those which have at least 5% CDS overlap with the selected model, were discarded. Scaffolds identified as composed of predominantly bacterial or organellar taxonomy (>10 gene models) were removed as assembly contaminants. Selected gene models form the GeneCatalog, which is available for further genome analysis and potential manual curation on PhycoCosm.

Functional annotations were assigned using InterProScan v5.57-90.0 and eggNOG-mapper v2.1.10 with multiple queried databases, namely Pfam v35.0, PANTHER v15.0, TIGRFAM v15.0, and EggNOG v5[105,120,121]. Organellar genomes were annotated by MFannot (http://megasun.bch.umontreal.ca/cgi-bin/mfannot/mfannotInterface.pl) and manually curated by homology searches. Organellar annotations were visualized by OGDraw[122]. MFannot and OGDraw were last accessed on November 15th, 2020. Heterotrophy indexes for annotated genomes were calculated based on KEGG marker genes[123]. In the case of Tara Oceans Gene Atlas (MATOU) unigenes, heterotrophy indexes were calculated based on all detectable KEGG orthologs in each station/depth (occurrence > 0).

## Metagenomic and metatranscriptomic read mapping and analysis

We analyzed raw reads from Tara Oceans, CalCOFI (NCOG), the Baltic Sea section of The Sorcerer II Global Ocean Sampling Expedition, Atlantic pole-to-pole, and Southern Ocean (CICLOPS) projects that mapped marine diversity globally[52,56,57,59,60] (Supplementary Data 3). Reads from these samples were mapped to genomic data masked by RepeatMasker v4.0.7 (www.repeatmasker.org) and processed by a pipeline consisting of SRA-Tools v2.10.9 (NCBI; ncbi.github.io/sra-tools), HISAT2 v2.2.1[124], SAMtools v1.11[125], BEDTools v2.26.0[126], and a custom read-filtering Python3.6 script (assign_reads2genomes.py available in the OSF repository). Briefly, read archives prefetched from NCBI SRA collection were mapped to combined repeat-masked assemblies using the splice-aware mapper HISAT2 and then filtered to remove secondary and low-quality reads and reads consisting of more than ~70% nucleotide repeats using a higher-order Markov model entropy filter (adapted from[127]). In the last step, reads mapping to multiple assemblies were also identified. Data were stored as BAM files, allowing downstream data analyses to be performed in Python 3 with standard libraries. Maps were generated using the Python Matplotlib toolkits' v1.2.1 basemap library.

For CalCOFI (NCOG) data, a metaT assembly was generated to account for sequences not covered by our strain and MAG genomic data. To date, 307 RNA samples were collected on quarterly CalCOFI cruises from 2014-2020 onto 0.22 μm Sterivex filters (Sigma-Aldrich). Following filtration, samples were immediately flash frozen in liquid nitrogen, then stored at −80 °C post-cruise. RNA was then extracted with the Macherey-Nagel NucleoMag RNA kit on an Eppendorf epMotion 5075TMX[128].

Poly-A selected cDNA from total RNA was generated with the SMART-Seq v4 Ultra Low Input RNA Kit for Sequencing (Takara Bio USA) which was then sheared with a Covaris ultrasonicator. The final sequencing library was then constructed with the NEB NEBNext Ultra II

DNA Library Kit and sequenced on three lanes of a NovaSeq 6000 with S4 flow cell (2×150 bp).

metaT assemblies were generated using the RNAseq Annotation Pipeline[54]. Briefly, the raw reads were trimmed for quality and adaptor removal. Ribosomal RNA (rRNA) sequences were removed with Ribopicker v0.4.3[129]. Trimmed and filtered reads were then used for assembly into contigs and abundances were quantified by mapping these reads to the assembly. Both assembly and read mapping were performed with CLC Bio Genomics Server v21.0.3. Gene prediction was performed with FragGeneScan v1.16[130] and rRNA removal was performed again. Predicted proteins were further filtered to remove those less than 10 amino acids long or with greater than or equal to 20% stop codons. Phaeocystales open reading frames (ORF) were identified via DIAMOND blastp against PhyloDB v1.076[54] and the Lineage Probability Index (LPI)[131]. Gene clusters were generated from the predicted proteins with MCL v14-137[132] with the inflation option (-I) set to 4 and scheme option (-scheme) set to 6. We used Self-Organizing Maps (SOM)[133] as a secondary clustering approach, reducing the complexity of the data into a handful of core transcriptional clusters that we could then explore in relation to environmental parameters[134]. To quantify the relationships between variable transcription in *Phaeocystis* (SOM clusters) and environmental gradients, we applied a Dirichlet-multinomial regression approach using the DirichletReg package[135]. Model fit calculated using the Akaike Information Criterion (AIC) identified which environmental variables best predicted the relative abundance of core transcriptional clusters.

Generalized additive models (GAMs) were calculated using the Mixed GAM Computation Vehicle package for R (mgcv version 1.9[136], R version 2023.09.0 + 463). Normalized read counts were smoothened against environmental variables from Ocean Gene Atlas v2.0[137] and diatom or dinoflagellate abundances taken from MATOU matrices (summed abundance of all unigenes with taxonomic classification "Bacillariophyceae" or "Dinophyceae"). The normalized read count data had an approximately normal distribution after log transformation (based on skewness ~0 and kurtosis ~3). The smoothing parameter was determined by the restricted ML method (method = "REML") with a maximum of 5 basis functions (k = 5). Only Tara Oceans records with chlorophyll a data were kept; records for TARA_085-SRF were removed as outliers (n = 239 samples). The fitting parameters included: gam(logtransformed_normreads ~ s(logChl_a) + s(Iron_5m, k = 5) + s(Nitrate_5m, k = 5) + s(log(Ammonium_5m)) + s(diatoms, k = 5) + s(dinophyceae, k = 5) + s(Temperature) + s(Distance_coast)). Statistics were also calculated individually for each independent variable. Species with low global abundance were omitted from the analysis. Data for large size fractions should be interpreted with caution due to the low number of samples with sufficient abundance data.

For the Arctic (subset of Atlantic pole-to-pole and Tara Arctic samples from the Norwegian, Greenland, Barents Sea, and West Siberian off-shore; Supplementary Data 3) and Southern Ocean biotopes (CICLOPS), metaT assemblies were generated using a modified RNAseq Annotation Pipeline[54]. Briefly, raw reads were adapter- and quality-trimmed using fastp v0.23.2 (trimmomatic option)[138,139]. rRNA-matched trimmed were removed using BBduk (https://jgi.doe.gov/data-and-tools/software-tools/bbtools/; rDNA databases PR2 v4.12.0[140], RFAM v14.1[141], SILVA v138[142]). Metatranscriptomes were assembled by biosample using MEGAHIT v1.2.9[143], and the longest contigs were retained via mmseqs2 release 14 clustering (0.95 sequence identity)[144]. Open reading frames were determined and translated by FragGeneScan[130] followed by annotation by InterProScan v5.57-90.0 using the protein domain database Pfam v35.0[105]. Phaeocystales ORFs were identified as above. To allow quantitative comparison of data from different projects, we generated MCL (-I 4 -scheme 7) gene clusters from the combined datasets of the Arctic, Northeast Pacific (CalCOFI), and Southern Ocean. First, ORF read counts were obtained by Bowtie2 v2.5. (--local mode, otherwise as

default)[145] using euphotic biosamples. The raw count matrix was then normalized by ORF length and *Phaeocystis* ORF sum per biosample (TPM), summed by MCL cluster, and filtered to remove clusters with mean TPM ≦10 in fewer than two biotopes. Where multiple Pfams were found per orthogroup, the original data row was divided into individual rows, one per Pfam. The expression metrics for these rows were inherited from the original row (i.e., not divided among Pfams). Statistical significance of differential expression of clusters and Pfams between biotopes was assessed by pairwise Mann-Whitney U tests, multiple comparison-adjusted by the Benjamini/Hochberg method using a strict 0.001 *p*-adj threshold ($n_{Arctic}$=75, $n_{SOC}$ = 30, $n_{CCE}$ = 223). Finally, for biotopes' ternary comparisons, mean TPM was normalized to per-row (orthogroup/Pfam) sums to reflect the proportion of recovered expression of each orthogroup/Pfam in each biotope. Additionally, differential expression was assessed on cluster raw counts normalized to library size (the three biotope sets differ substantially in their sequencing depths; abundances attributed to non-target taxa were aggregated) by ANCOM-BC2 v2.9.1[146] using pairwise comparisons of the three biotopes/groups of interest. We used a prevalence cutoff of 0.05 to avoid the removal of clusters exclusively present in the Southern Ocean biotope (with the smallest sampling size). Maps were generated using the Python Matplotlib toolkits' v1.2.1 basemap library.

Additionally, we analyzed unigene occurrences from the Tara Oceans MATOU v2 for correlation with associated environmental variables. All unigenes taxonomically annotated as Phaeocystaceae were included, except those lacking a good blastn hit (i.e., percent identity >70%, unigene coverage >50%; BLAST v2.13.0+) in our combined Phaeocystales nucleotide database used for biogeography. The unigenes' occurrences were summed per station and nominal depth, including depths designated as surface, deep chlorophyll maximum (DCM), mixed, or ZZZ, and then summed by Pfam annotation. Pfams with mean occurrence ≦$10^{-5}$ were removed. Environmental metadata were obtained from the Ocean Gene Atlas v2.0[137] and correlated with Pfam occurrence (n = 141 samples) using two-sided Spearman correlation, multiple comparison-adjusted by the Benjamini/Hochberg method.

### CalCOFI (NCOG) 18S-V4 rDNA abundances

Phaeocystales 18S-V4 rDNA abundances in the California Current were investigated with the NCOG dataset described in James et al.[52]. Here, 813 samples from the years 2014-2016 and 2018-2020 were collected from the near-surface (normally 10 m) and the subsurface chlorophyll maximum onto 0.22 μm Sterivex filters. Following filtration, samples were immediately flash frozen in liquid nitrogen, then stored at −80 °C post-cruise. DNA was extracted with the Macherey-Nagel NucleoMag Plant kit on an Eppendorf epMotion 5075TMX and assessed on a 1.8% agarose gel. At the start of DNA extraction (addition of lysis buffer), 1.74 to 3.78 ng of *Schizosaccharomyces pombe* genomic DNA was added to each sample as an internal standard[147].

Amplicon libraries were constructed via a one-step PCR using the TruFi DNA Polymerase PCR kit and the V4F (5'-CCA GCA SCY GCG GTA ATT CC-3') and V4RB (5'-CCA GCA SCY GCG GTA ATT CC-3') primer set[148]. Each reaction was performed with an initial denaturing step at 95 °C for 1 minute followed by 30 cycles of 95 °C for 15 seconds, 56 °C for 15 seconds, and 72 °C for 30 seconds. 2.5 μL of each PCR reaction was run on a 1.8% agarose gel to confirm amplification, then PCR products were purified with Beckman Coulter AMPure XP beads following the manufacturer's instructions. PCR quantification was performed in duplicate using the Invitrogen Quant-iT PicoGreen dsDNA Assay kit. Samples were then pooled in equal proportions into seven separate pools followed by another 0.8× AMPure XP bead purification on the final pool. DNA quality of the final pool was evaluated on an Agilent 2200 TapeStation and quantification was performed with the

Qubit HS dsDNA kit. Sequencing was performed on Illumina MiSeq (2×300 bp) at the University of California, Davis Sequence Core.

Amplicons were generated and analyzed with QIIME2 v2019.10[149]. Briefly, paired-end reads were trimmed to remove adapter and primer sequences with cutadapt[150]. Trimmed reads were then denoised with DADA2 to produce amplicon sequence variants (ASVs; maxEE = 2, chimera-method = "pooled"). Each MiSeq run was denoised with DADA2 separately to account for different error profiles in each run, then merged. Taxonomic annotation of ASVs was performed with q2-feature-classifier using the naïve bayes classifier and the PR² database (v4.13.0)[140,151].

*Phaeocystales* 18S copies per Liter were estimated as described by Lin et al.[147]. Within each sample, reads were divided by the ratio of *S. pombe* reads and the number of *S. pombe* rRNA copies added. The total number of copies was then normalized to the volume filtered for each sample to estimate copies $L^{-1}$.

### *P. globosa* transcriptome library preparation, sequencing, and quantification

Total RNA was extracted from the filters using the NucleoMag RNA kit (Macherey-Nagel, Düren, Germany). rRNA was depleted using Ribo-Zero Magnetic kit (Illumina, La Jolla, USA) with a modified Removal Solution consisting of plant, bacterial, and human/mouse/rat solutions (2:1:1 ratio). cDNA was synthesized by the Ovation RNA-Seq System V2 (Tecan, Redwood City, USA), which was then fragmented to the target size of 400 bp using the Covaris E210 focused ultrasonicator. Libraries were prepared using the Ovation Ultralow V2 system (Tecan) and purified by AMPure XP beads (Beckman Coulter Life Sciences, Brea, USA). Libraries were subjected to paired-end 2×150 bp sequencing on a NovaSeq 6000 instrument (Illumina) to an average of 24 million reads per library. Raw RNAseq reads (available at the JGI genome portal under the Phaglo1 accession) were mapped to the repeat-masked genome assembly using the splice-aware read aligner HISAT2 v2.2.1[124]. Read mapping counts were extracted using SAMtools-1.16.1 and BED-Tools v2.30.0[125,126] and normalized to transcript length and library size (TPM). The transcriptomic data were primarily generated to allow efficient gene prediction and consisted of mostly single biological replicates. Therefore, differential expression was performed with biological functions as in ref. 55. Briefly, TPM values were pooled for genes with identical inferred KEGG orthologs or Pfam domains, and these values were compared between various conditions using Analysis of Sequence Counts, ASC v0.1.4, a Bayesian posterior probability method[152]. The differential expression of genes associated with higher mitochondrial-to-plastid transcription was also assessed by ANCOM-BC2[146] (default parameters).

### Repetitive elements

Reference genomes were first analyzed using Tandem Repeats Finder[153] (TRF) version 4.04 (--maxPeriod 10) to mask tandem repeat regions of at least 100 bp. We then used the REPET v3.0 package to annotate dispersed repetitive elements. Briefly, we launched TEdenovo[154] to generate a library of consensus sequences representative of repetitive elements in each genome assembly. Each library was classified using PASTEC[153] and sequences classified as simple repeats were removed. Each library was then used to select the consensus sequences with at least one full length copy using TEannot[155]. The final libraries were used to annotate the respective genomes using TEannot again. The consensus sequences that remained unclassified with PASTEC were searched for ORFs encoding proteins with a minimum length of 200 aa. For each species, these ORFs were clustered at 40 % identity using MMseqs2[144] and representative proteins from each cluster were scanned for homology with known structures using the HH-suite v3.3.0 as described below. Simple sequence repeats were separately searched with TRF with two sets of parameters: 2 10 10 80

10 24 2000 (soft) and 2 3 5 80 10 20 2000 (aggressive) and with the sDUST v0.1 algorithm[156].

## Search for endogenous virophages/PLVs and NCLDVs

Reference genomes were searched by NCLDV and virophage proteomes from UniProt, NCLDV HMM profiles from ref. 157 and viral metagenomes HMM profiles from IMG_VR_2020-10-12_5.1[158] (hosted at the JGI Genome portal). We also inferred a taxonomic origin for all predicted proteins using the "taxonomy" module of MMseqs2[144] against the UniRef90 database. The genomic positions of candidate viral proteins and loci were merged when distant less than 10kb, and the corresponding fasta sequences were screened for hallmarks of NCLDV, virophages, and PLVs, including their size and the presence of core genes. ViralRecall v1[159] was also launched on the Phaant1 and Phaglo1 genomes and the output used as a complementary source of information. Endogenous viral ORFs were predicted using Prokka 1.14[160]. The structure-based annotations were obtained using HH-suite v3.3.0[161] with UniRef30 sequence database and PDB70 structure database and the sequence homology annotations were obtained using BLASTP against the GenBank nr database (accessed 12/1/2023).

## Phylogenetic and phylogenomic analyses

Homologs of sequences of interest were searched in NCBI GenBank-nr, EukProt2[162], JGI-genome (genome.jgi.doe.gov), and recently published viral[81,163] databases using DIAMOND v2.0.14.152[97]. To infer phylogenies, datasets were aligned by MAFFT v7.407[164] using the L-INS-i refinement and a maximum of 1000 iterations, followed by trimAl v1.4[165] trimming of sites with >70% gaps (-gt 0.3). ML trees were inferred by IQ-TREE v1.6.12[166] using the GTR + F + I model (nucleotide) or Posterior Mean Site Frequency (PMSF)[167] model with a C20 guide tree (protein) and employing 1,000 ultra-fast bootstrap replicates and 1,000 SH-aLRT replicates. For the PLV MCP phylogeny, protein sequences were aligned with PROMALS3D[168] using the Paramecium bursaria chlorella virus type 1 as model (PDB: 5TIP, https://www.rcsb.org/structure/5TIP). Poorly conserved positions were trimmed by trimAl, and the phylogenetic tree was constructed by IQ-TREE as above. For the NCLDV MCP, protein sequences were aligned and processed as above. Recombination events between endogenous viruses were detected using RDP4 v4.101[169]. The phylogenetic analysis of rhodopsins was performed with the alignments from Rozenberg et al.[170]. *Phaeocystis* sequences were added using the --add option of MAFFT v7.511 (--keeplength), followed by IQ-TREE ML tree inference using the WAG + F + R3 model and employing 1,000 ultra-fast bootstrap replicates.

For phylogenomic analyses, single-gene datasets were processed using PhyloFisher v1.1.0[171], and paralogs were removed manually. Following concatenation into a multi-gene supermatrix (17 longest-gene matrix 14,953 sites; 240-gene matrix originally 71,716 sites, then 6,000 fastest-evolving sites removed by PhyloFisher for the phylogenomic tree), multi-gene ML trees were reconstructed using the PMSF model with a C20 guide tree (IQ-TREE)[166,167]. Ultra-fast bootstrap (up to 1,000 iterations) and SH-aLRT (1,000 replicates) were calculated as branch support. Fast-evolving sites were removed step-wise by PhyloFisher, allowing a consistency check of the resulting topologies. Timetrees were calculated using the 17-protein alignment with the least complete and taxonomically redundant accessions and sites with 20% highest variability removed by tiger v2.0[172] (-b 10 -exc 9,10; 50 sequences, 10,766 sites remaining). BEAST v2.2.1[173] was used to infer the divergence times with the WAG + I + G site model (3 gamma rates, 0.22 invariant proportion) and the Relaxed Log Normal clock model; log normal age priors on two nodes[20] (220 ± 4 Mya on Coccolithophora, 65 ± 2 Mya on *Calcidiscus × Coccolithus*); and birth rate determined by the Calibrated Yule model; other parameters estimated by the algorithm. Three Markov chain Monte Carlo (MCMC) chains were run for 127 million generations, sampling every 1,000 generations. The runs

were inspected for convergence of topologies, log-likelihoods and parameter values in Tracer v1.7.1[173]. First 25 million trees were discarded as burn-in, and the remaining trees were used by TreeAnnotator v2.6.4[173] to build the consensus tree and to calculate the posterior probabilities of each node. Mash distance analysis (MASH-ANI v2.3), which approximates average nucleotide identity[174], was used to determine nuclear and organellar genome divergence.

Pfam enrichment (evolutionary distance-calibrated fast-evolving gene family enrichment) was performed using InterProScan annotations and CAFE v4.2.1[175]. The dataset included protein models inferred for algal lineage representatives with genomic data; only non-overlapping gene models from *Phaeocystis* were used (no isoforms). The PSC dataset was compiled from the proteomes of four closely related MAGs (AOS_82_MAG_00142, MED_95_MAG_00439, PSE_93_MAG_00224, PSW_86_MAG_00287; PSC1 in Supplementary Data 1) by clustering them with MMseqs2 v14 to remove redundancy (--min-seq-id 0.95 -c 0.8, a threshold chosen based on BUSCO duplication rate of the resulting representative gene set)[99,144]. First, protein domains identified were counted in the predicted proteomes of selected representatives of algal lineages with complete genome assemblies. Short protein domains (<30 aa) were skipped. The phylogenetic distances of the analyzed algae were obtained from the multi-gene tree and recomputed to ultrametric using r8s v1.80[176]. The birth-death parameters $\lambda$ and $\mu$ were estimated globally, and the best model to account for genome assembly error was determined by the CAFE run. Finally, gene family evolution rates and their significance were calculated and, for fast evolving families, associated KOG biological processes were assigned.

Pfam abundances at stations were obtained either directly from the MATOU v2 atlas[137] (for Tara Oceans and Tara Arctic), or adopting a read-mapping method using our combined assembly (for stations additionally including the P2P and CICLOPS collections). For ISIPs, dsyB, Alma1, and methionine synthases, a reference query protein was used to find a broader set of homologs using PSI-BLAST v2.13.0 +, these were aligned using muscle5.1[177], trimmed to remove sites with >70% gaps[165], and used to build HMM profiles to search the proteomes of Phaant1, Phacord1, Phaglo1, and PSC using HMMer v3.3.2[178]. Additionally (for tubulin, nitrite/sulfite reductase PTHR32439, carbonic anhydrase PF00484 and PTHR18952, vacuolar ion transporter VIT/Ccc1 PF01988, xanthorhodopsins PF01036, ferredoxin PF00111, and flavodoxin PF00258), existing Pfam/PANTHER annotation coordinates were used to extract the corresponding protein sequences from the above Phaeocystales proteomes (ferredoxin, flavodoxin as in ref. 13). Lastly, a subset of the phylogenomic markers (58 genes with no paralogs found in the search phase; available in OSF repository: https://osf.io/vka93) was used for metaG normalization; these single-copy conserved orthologs were extracted from the phylogenomic datasets. Next, TBLASTN (-evalue 1E-3; v2.13.0+) identified the coordinates of the Phaeocystales protein hits in the compiled Phaeocystales genomic data (repeat-masked, used for read mapping). The coordinates were manually filtered to remove off-targets and used to extract read mapping information from the above-generated BAM files. These read counts were then processed with Python3 code, including normalization to the total *Phaeocystis* read count in each biosample. Their correlation with iron levels were analyzed using two-sided Spearman correlation (n = 49 Southern/Arctic Ocean and n = 82 other ocean samples).

Orthogroups were found using OrthoFinder v2.3.11[179]. To detect horizontal gene transfers, complete proteomes of *Phaeocystis* spp. (Phaant1, Phacord1, and Phaglo1; 100,898 "ingroup" or "query" sequences) and Haptophyta+Centroplasthelida (486,407 sequences considered closest sister to Phaeocystales) and up to 100 DIAMOND[97] blastp hits from NCBI-nr, EukProt, EggNOG, and OM-RGC retrieved (695,920 "outgroup" sequences) with *Phaeocystis* sequences as queries. In total, ~1.25 million sequences from all major eukaryotic,

prokaryotic and viral clades, were grouped by OrthoFinder. The orthologous groups (orthogroups, OGs) with at least one *Phaeocystis* sequence (14,607 OGs larger than 2) were aligned with MUSCLE v5.1[177] (-align mode for small OGs, -super5 for large), trimmed by trimAl v1.4.rev15[165] (-gt 0.3), and their unrooted phylogenies were inferred using FastTree v2.1.8[180] under the WAG model. Starting from each *Phaeocystis* query, a custom tree-walking algorithm adapted from ref. 181 evaluated the origin of sibling sequences in the current highly supported (FastTree bootstrap>0.85) monophyletic clade. The query's evolutionary origin was determined when one of the stopping criteria was met. A clade was: 1) "ancestral" when 5 sister sequences were found; 2) "ancestral-HGT" when sister sequences and at least 3 outgroup sequences from one major lineage were found; 3) "single-HGT" or "multi-HGT" when only ingroup and at least 3 outgroup sequences from major lineages were found (i.e. no haptistan sisters), and depending on the large taxonomic composition of the clade (a single or multiple lineages, respectively); 4) "ingroup-only" if exclusively ingroup sequences were found in the tree. The clade was considered ingroup-monophyletic if all ingroup and sister sequences were nested within outgroup sequences (as opposed to branching next to them), assuming the clade's ancestral node is the root. This was important to infer the direction of gene transfers, which is otherwise difficult using unrooted topologies. Using ingroup monophyly, we could confidently refine HGT events where Haptophyta/Phaeocystales acted as acceptors rather than donors of genes. HGT genes were not found on contigs significantly shorter than "ancestral" genes (Mann-Whitney U test, Vargha-Delaney A effect size; Phaant1: $U = 6.2e + 05$, $p = 0.937$, $A = 0.559$; Phaglo1: $U = 5.1e + 05$, $p = 0.615$, $A = 0.51$; Phacord1: $U = 8.5e + 05$, $p = 0.357$, $A = 0.49$;), corroborating that they are not artifacts of uncaught genome assembly contamination. When tested for enrichment in HGT clades, lineages inferred as participating in HGTs were more abundant in HGT clades than expected from random distribution based on the respective trees' taxonomic composition (Benjamini/Hochberg-adjusted one-sided Wilcoxon test, results in Supplementary Data 8).

## Reporting summary

Further information on research design is available in the Nature Portfolio Reporting Summary linked to this article.

## Data availability

Sequence read archives for Southern Ocean (CICLOPS) data were deposited at NCBI GenBank under BioProject PRJNA890306; sequence read archives for NOAA CalCOFI Ocean Genomics (NCOG) Program polyA-enriched libraries were deposited under BioProject PRJNA1088233;. The genome assemblies and annotations are available at the DOE Joint Genome Institute portal PhycoCosm[38] and have been deposited in DDBJ/ENA/GenBank with the following URLs: Phaant1: https://phycocosm.jgi.doe.gov/Phaant1; https://www.ncbi.nlm.nih.gov/bioproject/PRJNA34537/ Phacord1: https://phycocosm.jgi.doe.gov/Phacord1/; https://www.ncbi.nlm.nih.gov/bioproject/PRJNA534932/ Phaglo1: https://phycocosm.jgi.doe.gov/Phaglo1/; https://www.ncbi.nlm.nih.gov/bioproject/PRJNA265550/ Other processed data (assemblies, annotations, and phylogenetic data) are available at the OSF repository: https://osf.io/vka93 All previously published data used here are listed in Supplementary Data 3, sheet Processed reads. Source data are provided with this paper.

## Code availability

All code for data cleaning and analysis associated with the study is available at the OSF repository: https://osf.io/vka93.

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

## Acknowledgements

This project is funded by National Oceanic and Atmospheric Administration grants NA15OAR4320071 and NA19NOS4780181 (to AEA), the National Science Foundation (NSF OCE-1756884 and NSF OCE-2224726 to AEA, NSF IOS-1557928 to ABK, NSF OPP-1643684 to MAS), and the Simons Collaboration on Principles of Microbial Ecosystems (PriME) (Grant ID: 970820 to AEA). The work (proposal: 10.46936/10.25585/60001426) conducted by the U.S. Department of Energy Joint Genome Institute (https://ror.org/04xm1d337), a DOE Office of Science User Facility, is supported by the Office of Science of the U.S. Department of Energy operated under Contract No. DE-AC02-05CH11231. ZF received funding from Fulbright Slovak Republic. MO received funding from the Czech Science Foundation (Grant ID: 23-06203S). Computational resources were provided via support by the Ministry of Education, Youth and Sports of the Czech Republic through e-INFRA CZ (ID:90254). FM (via affiliation to URGI) benefits from the support of Saclay Plant Sciences-SPS (ANR-17-EUR-0007) and the PlantBioinfoPF platform. JD and CJ were supported by CNRS and ATIP-Avenir program funding and by the European Union (GA#101059915 - BIOcean5D). Views and opinions expressed are, however, those of the author(s) only and do not necessarily reflect those of the European Union. Neither the European Union nor the granting authority can be held responsible for them. The authors would like to thank three reviewers for their constructive feedback, and Freya Hammar and Anna Oborníková for their artistic input on the featured image.

## Author contributions

AEA conceived the project and provided overall project leadership. ZF co-supervised the work, analyzed the data, and wrote the manuscript; K.R.A. provided *P. antarctica* strain material; C.P.D.B. performed viral experiments and provided the North Sea *P. globosa* Pg-G(A) strain material; G.R.D. and M.A.S. provided Southern Ocean samples; M.E.F. provided colony EST data; A.B.K. provided metal experimental samples; C.dV and I.P. performed genome sizing experiments; HZ provided laboratory work; K.R.A., K.B., D.M.G., I.V.G., R.D.H., A.L.H., J.W.J. and J.S. performed genome assembly; K.B., D.M.G., I.V.G., R.D.H. and J.S. performed gene annotation; M.M.B. helped analyzing *P. pouchetii* data; F.M. and H.T. analyzed endogenous virus sequences; L.D.H.E., B.W.M. and I.T.P. annotated the transporters; M.K., D.T.C. and K.Z. provided metabolomic models; Z.F., M.O., T.S. and A.M.G.N.V. performed horizontal gene transfer analysis; C.J. and J.D. contributed to the oceanic domain functional enrichment analysis; R.H.L., C.C.J. and P.Ven analyzed the CCE-NCOG data; Z.F. and PVec analyzed *P. globosa* RNAseq data; A.E.A. provided supervision and experimental design. A.E.A., K.R.A., M.M.B.,

C.P.D.B. and R.H.L. contributed to the manuscript. All authors approved the submitted and revised versions of the manuscript.

## Competing interests

The authors declare no competing interests.

## Additional information

[1]Scripps Institution of Oceanography, University of California San Diego, La Jolla, CA, USA. [2]Microbial and Environmental Genomics, J. Craig Venter Institute, La Jolla, CA, USA. [3]University of South Bohemia, České Budějovice, Czech Republic. [4]Stanford University, Department of Earth System Science, Stanford, CA, USA. [5]United States Department of Energy Joint Genome Institute, Lawrence Berkeley National Laboratory, Berkeley, CA, USA. [6]University of South Florida, St. Petersburg, FL, USA. [7]NIOZ – Royal Netherlands Institute for Sea Research, Den Burg, The Netherlands. [8]Department of Freshwater and Marine Ecology, Institute for Biodiversity and Ecosystem Dynamics (IBED), University of Amsterdam, Amsterdam, The Netherlands. [9]Cell and Plant Physiology Laboratory, CNRS, CEA, INRAE, IRIG, Université Grenoble Alpes, 38054 Grenoble, France. [10]Station Biologique de Roscoff, CNRS / Sorbonne Université, Roscoff, France. [11]Hollings Marine Laboratory, College of Charleston, Charleston, SC, USA. [12]School of Natural Sciences, Macquarie University, Sydney, Australia. [13]ARC Centre of Excellence in Synthetic Biology, Macquarie University, Sydney, Australia. [14]Skidaway Institute of Oceanography, University of Georgia, Savannah, GA, USA. [15]Department of Plant and Microbial Biology, University of California Berkeley, Berkeley, CA, USA. [16]Genome Sequencing Center, HudsonAlpha Institute for Biotechnology, Huntsville, AL, USA. [17]University of Southern California, Los Angeles, CA, USA. [18]Department of Pediatrics, University of California San Diego, La Jolla, CA, USA. [19]Department of Earth and Environmental Sciences, Rutgers University – Newark, Newark, NJ, USA. [20]Université Paris-Saclay, INRAE, URGI, 78026 Versailles, France. [21]Institut de Biologie de l'École Normale Supérieure (IBENS), CNRS, Paris, UK. [22]Biology Centre of the Czech Academy of Sciences, Institute of Parasitology, České Budějovice, Czech Republic. [23]Woods Hole Oceanographic Institution, Woods Hole, MA, USA. [24]Université Paris-Saclay, INRAE, Institute of Plant Sciences Paris-Saclay (IPS2), Gif sur Yvette, France. [25]Department of Bioengineering, University of California San Diego, La Jolla, CA, USA. [26]Center for Microbiome Innovation, University of California San Diego, La Jolla, CA, USA. [27]Program in Materials Science and Engineering, University of California San Diego, La Jolla, CA, USA. [28]Present address: European Molecular Biology Laboratory, 69117 Heidelberg, Germany. [29]Present address: Université Paris-Saclay, INRAE, AgroParisTech, Institute Jean-Pierre Bourgin for Plant Sciences (IJPB), 78000 Versailles, France. ✉e-mail: aallen@jcvi.org

