## [Transparent Peer Review file · Nature Communications]

Genome-resolved biogeography of Phaeocystales, cosmopolitan bloom-forming algae

Corresponding Author: Professor Andrew Allen

Version 0:

Reviewer comments:

Reviewer #1

(Remarks to the Author)

The manuscript titled " Genome-resolved biogeography of Phaeocystales, cosmopolitan bloom-forming microalgae" explores the global distribution and ecological adaptations of the genus Phaeocystis through genomic, metagenomic, and transcriptomic analyses and comparisons. The authors utilize large-scale in situ data from expeditions and in vitro culture data from multiple Phaeocystis strains. This study is distinguished by its integrative approach.

The authors utilized genomic data from thirteen strains of five Phaeocystis species. These data were assembled and annotated, providing a solid foundation for phylogenetic and biogeographic analysis. The analyses also include 21 genomes assembled from metagenomes (MAGs) collected during various expeditions. These MAGs expand the taxonomic sampling and include uncultivated species.

Transcriptomic data were generated under controlled conditions to study the responses of Phaeocystis to different levels of nutrients and light. These experimental data provide valuable insights into the regulatory mechanisms and physiological adaptations of Phaeocystis, complementing environmental observations.

This manuscript presents an impressive accumulation of analyses and figures; however, for almost all analyses, the conclusions are very general, vague, or the authors fail to reach definitive conclusions. This sentiment is evident from the abstract, where it is clear what the authors have accomplished and the potential value of the new data. However, it is difficult to discern what specific answers the authors provide regarding a better understanding of the biogeochemical cycles of carbon and sulfur.

It would undoubtedly be more useful to explore fewer questions but in greater depth. For instance, the analyses on gene function repertoires, gene expansions, and horizontal gene transfer (HGT) are very interesting. Since the authors use transcriptomic data, we expect an analysis of gene expression levels as well as an integration of the physicochemical environmental data. I encourage the authors to better exploit the immense quality of the data to draw more synthetic and clear conclusions on the explanation of biogeography and/or major biogeochemical and/or evolutionary functions.

The authors analyze and compare relative abundances based on sampling, which results in compositional data. Numerous articles have been published in recent years highlighting the many inherent biases in compositional data, which can lead to erroneous conclusions when compared. There are statistical methods available to correct or mitigate these biases. For example, see doi.org/10.1038/s41467-020-17041-7. Therefore, I wonder if, for some analyses at least, validation of the absence of bias would be necessary.

Many figures are much too small.

The various cultures sequenced in this study were done so using very different sequencing and assembly protocols. Although I do not think this affects the main conclusions, these technical differences can lead to varying quality levels across genomes and, consequently, different biases, particularly in repetitive regions, gene family expansions, etc. The authors should comment on this point.

The gene prediction method is a crucial and very delicate step for environmental eukaryotic genomes. The authors simply cite "phytozome and mycozome pipelines." References 9 and 10 indicated are not methodological pipelines; they are genome databases. The authors must clearly explain how the exon-intron structures of nuclear genes are predicted. Are

introns present (the word is completely absent from the main and supplementary text)? Additionally, I do not see any statistics on gene and protein sizes anywhere.

The analysis of viral sequence integrations does not incorporate a significant collection from the new clade of mirusvirus (doi: 10.1038/s41586-023-05962-4), nor the following collection for NCLDVs: doi.org/10.1038/s41586-020-2924-2. The in silico metabolic reconstruction is very interesting. Can we associate these differences in carbon and nitrogen sources with environmental parameters and evolutionary proximities between lineages or species?

Details:

Abstract:

"highest-contiguity" is relative and subjective. It's better to use more precise terms.

"fine-tuned preferences" – Use another term than fine-tuned, as it can sometimes be misinterpreted (e.g., intelligent design).

Text:

"The gene repertoires of three annotated Phaeocystis reference genomes reflect a unique strategy of gene expansion among algae: repetitive elements, horizontal gene transfer, and full-length endogenous virus insertions." Do we have enough sequenced algae (prokaryotes and eukaryotes) to support this statement?

The statement "Phaeocystales thus emerge as an ecologically versatile group with diverse adaptations to biotic and abiotic stressors" is very vague and general. It is hard to imagine many groups for which these conclusions would not be true.

Line 153: The placement of references 46 and 47 in the sentence creates ambiguity about their role. I suggest adding citations of examples from the numerous references in the literature on the biogeographic distribution of eukaryotic phytoplankton based on barcodes in the first part of the sentence. Furthermore, biogeographic data on Phaeocystis from genome data, particularly MAGs, already exists. Reference 29, among others, provides numerous maps (or links towards maps) of this lineage and should therefore be discussed in the introduction at this point in the manuscript.

Lines 203 and the subsequent discussion are very interesting. Have the authors analyzed the ratio of genomic abundance between the nucleus and organelles? This analysis could reveal variations in the number of organelles per cell, providing significant insights into the transcriptomic analysis and potentially highlighting important aspects of cellular regulation and adaptation.

Figure 1A: The bar plots for the unigene collection represent the composition of the collections, not the communities. To enable a useful comparison between the 'maredat' and 'unigene collections' bar plots, the figures should represent the same type of information.

Additionally, the violin plots in Figures 1A and 1B display negative values, which is clearly an artifact of the representation. This issue must be corrected to ensure the accuracy and clarity of the visual data presentation.

Figure 1h: A significant zoom is required to discern the information in this figure, but at this level of magnification, the image becomes blurry. Improving the resolution or providing a more focused view would enhance the clarity and usefulness of this figure.

Figure 5 Legend: It is unclear how the dataset could originate from reference 56 as indicated. Please verify the source of the dataset and correct the reference if necessary.

More methodological details on the analysis of environmental drivers using Generalized Additive Models (GAM) need to be provided. This should include information on the specific variables considered, the model fitting process, and any validation or testing procedures employed to ensure the robustness of the analysis

Reviewer #2

(Remarks to the Author)

This is a solid seminal work bringing together a wealth of genomic and metagenomic information available on Phaeocystis. The authors review and refine the phylogeny and biogeography of this globally significant genus of phytoplankton.

The authors conducted a detailed exploration of the global biogeography and adaptive strategies of Phaeocystis to variations in nutrient availability and form, temperature and environmental preferences through the genomes of several significant cultured species of Phaeocystis. They did this in combination with MAGs from several global expeditions to accommodate uncultured species. The data also provided cues on the varying levels of heterotrophy, metabolic flexibility and trophic strategies between the species/strains. Additionally, the authors also compared viral footprint integration between the Phaeocystis species and explored the interactions between virus and host through transcriptomic data.

We found the study to be novel, comprehensive and informative, contributing a significant amount of new data and analyses towards a better understanding of Phaeocystis. The study investigated the adaptive mechanisms and strategies of Phaeocystis, which will be key to understanding and predicting future shifts in Phaeocystis distribution and their contribution to global nutrient cycling, carbon sequestration, etc. We found the synthesis of results and findings around mixotrophy and

strategies to circumvent iron and vitamin B12 limitations particularly exciting.

The MS is clear, logical, easy to follow. We found it was very heavy on acronyms though which made it occasionally hard to follow. We suggest that the authors compile a list of acronyms used in the MS and their definitions on a single page that can be referred to on the first page of supplementary methods, for easier reading.

Specific comments:

Title/abstract:

It might help the reader if the authors explained upfront that Phaeocystales consisted of a single known genus plus a distantly related environmental sister clade (which would be a separate genus, based on genetic distance). At first we found it confusing that the title mentioned an order, but all the results seemed to talk of a single genus.

L107: We could not find MMETSP defined anywhere in the main text or supplementary materials.

L151: The two sentences seem to be contradictory to each other?

L255: We could not find dinoflagellates indicated in the text in Fig. 5. We suppose dinoflagellates are categorized as alveolates, but other subdivisions of alveolates also exist. If only dinoflagellates are found or classified as alveolates in this case, then it should be indicated explicitly in the diagram.

Supplementary Results (Endogenous viruses section): PgV was defined, but what is PgVV (also in Extended Data Fig. 8)?

Reviewer #3

(Remarks to the Author)

"I co-reviewed this manuscript with one of the reviewers who provided the listed reports. This is part of the Nature Communications initiative to facilitate training in peer review and to provide appropriate recognition for Early Career Researchers who co-review manuscripts."

Version 1:

Reviewer comments:

Reviewer #1

(Remarks to the Author)

I thank the authors for seriously considering the remarks and questions provided.

Overall, I remain of the opinion that this manuscript presents significant data and an impressive effort involving a large number of analyses.

However, the text often remains rather superficial and does not sufficiently highlight the quality of the data and analyses. A clear message does not emerge, which is exemplified by the abstract and conclusion, where the text remains fairly general.

Additionally, certain details in the results have been modified without explanation. While I understand that these could be drafting errors, the authors should provide justification for these changes, as they sometimes affect the results.

For example:

- Supp Line 601: Ammonium and nitrate have been reversed.
- Supp Line 601: "Dinoflagellate" has been removed.
- Supp Line 615: "Silicate" has been removed.

Some additional details:

- Supp (track change), line 881 : "Notably, however, flavodoxin was significantly downregulated in iron-880 replete conditions, in line with other results" : Add reference
- Abstract : "N50 >30,000." : missing unit

And my comments on your responses:

- "We expanded the conclusion on l. 189: Phaeocystales thus emerge as a ubiquitous component of global pico/nanophytoplankton, with apparent lineage-specific preferences."

The sentence remains very general; it is already well known that Phaeocystales is ubiquitous. Adding 'with apparent lineage-specific preferences' does not provide any additional insight. If different lineages exist, it is self-evident that they would exhibit differences; this is tautological.

- We newly conclude on l. 227: "We hypothesize that variable rates of mixotrophy and mitochondrial transcription contribute to this flexibility, and affect ecological niche partitioning between Phaeocystis lineages."

I agree that this sentence provides clarification.

- To make the narrative more cohesive, we revisited our analysis of horizontal gene transfers in more detail and added the following to Supplementary Results 6: "A total of 183 horizontal gene transfer (HGT) events were recorded, passing our most stringent criteria (see Supplementary Methods 8).

This new paragraph is indeed very interesting from an evolutionary, functional, and ecological perspective.

- We appreciate the raised concern and acknowledge that statistical analyses of compositional data are prone to biases, particularly in the absence of quantitative standards. For the metatranscriptomic and metagenomic analyses we used TPM normalization, which is commonly used when quantitative standards are not available (Cohen et al., 2022; doi: 10.3389/fmars.2022.867007).

A TPM normalization does indeed allow for normalization by transcript length and the number of sequenced reads, enabling the comparison of gene expression for transcripts of varying lengths across multiple samples. However, this method does not address compositional biases; it maintains them. A compositional bias occurs when the relative proportions of transcripts within a sample influence the results. This type of bias is particularly pronounced in cases where:

i) A high expression of a few transcripts (e.g., highly abundant genes) dominates the majority of reads.
ii) Variations in the relative proportions of transcripts between samples lead to distortions in the evaluation of less abundant genes.

If the authors are unable to use methods that avoid these biases or estimate their impact, they should, at a minimum, temper certain conclusions about expression comparisons or explain why potential biases would not affect their conclusions.

- We adjusted the figures and believe they comply with the Nature Publishing Groups guides for preparing figures, including font size (5-7pt).

In the PDF I received, some figures appear blurry when zooming in, for example, Extended 1E, Extended 4A, and Extended 8E.

- We added the following narrative to Supplementary Results 1 to reflect on these problems: "... likely stems from the limitations of short read-only data assembly. Indeed, according to GenomeScope 2.0 estimates, substantial portions of the data represent short repeats (37-69%; not shown). Due to these technical problems and limitations, repetitive regions remain difficult to assemble and quantify in *Phaeocystis* spp., along with recently duplicated genes they might flank. The varying levels of assembly quality introduce biases in coding and non-coding repeats, which is why we did not attempt to quantify gene family expansion in the fragmented assemblies."

Ok

- "The two annotation pipelines employ different intron-aware gene-finding worker algorithms, but fundamentally use a combination of homology-based modeling, ab initio predictions, and RNA/transcriptome evidence methods. Filtering combines different support metrics to select an appropriate single model to represent each locus. "

The authors provide very little technical information. One of the key principles of a publication is the reproducibility of results and data, to the greatest extent possible. Here, I do not see how this could be achieved without knowing the tools used, the specific computational codes, and the options or parameters applied.

- Response: As for the first reference, the dataset does not contain Yaravirus-like or *Pleurochrysis* sp. endemic viruses 1a and 2, since, as we elaborate in Supplementary Results 2., the key phylogenetic markers used in the study (DNApolB, RNAPa, RNAPb, TFIIS) are missing in *Phaeocystis* endogenous NCLDV and other Yaravirus-like viruses. The study therefore does not provide a suitable phylogenetic context for our endogenous NCLDV (the closest relatives from the dataset would be Asfaviridae with a relatively long branch). The second reference focuses on Mimiviridae, endogenous NCLDV only distantly related to the *Phaeocystis* NCLDV described in our manuscript. Nevertheless, we updated the phylogeny in Extended Data Fig S8B and added references to these recent works to account for the development in the NCLDV field.

OK

- Response : We developed these preliminary models to address specific questions, i.e., to predict the fluxes/relative fluxes in different conditions and for estimating mitochondrial/plastid flux ratios. These questions did not necessitate manual curation of the gene-reaction association, which is a time-consuming process, but as a result they do not allow linking fluxes with any gene-related data (e.g. transcriptomic data) or for finding marker genes for a trophic mode of these species, and can lead to erroneous outcomes. Based on our experience with *Cylindrotheca* (Kumar et al., 2024; doi:

10.1126/sciadv.ado2623), curating these models could take up to 3-4 months, with an additional 3-4 weeks to analyze the model predictions and validate them with environmental data. While these results are surely interesting, we would prefer to address them in a future work; perhaps with a specific focus on mixotrophy in nanoflagellates.

OK

- We replaced "three [genomes] with the highest contiguity" with "three with N50 >30,000", and replaced "fine-tuned" with "lineage-specific".

OK (but the unit is missing for N50)

- Response: Rephrased to a non-comparative statement:

"The gene repertoires of three annotated *Phaeocystis* reference genomes reflect an adaptation strategy through gene expansion: repetitive elements, horizontal gene transfer, and full-length endogenous virus insertions."

OK

- Response: That is probably true, but many of the phenomena we analyze (mixotrophy, endogenous viruses, functional gene family expansion) are emergent topics in algal genomics, and we anticipate their importance will grow with more experimental data. This sentence merely concludes the abstract, and we see it as a teaser to raise interest in the rest of the article.

I understand the authors' perspective. However, I find that this sentence, being overly general, undermines the quality of the analyses presented in this manuscript.

- "While reports on their global biogeography exist (REFs), they are based on amplicons or partial metagenomes, and do not elaborate on gene-level adaptation."

I see, but 'partial metagenomes' should be replaced with 'partially assembled genomes' or an equivalent term because "partial metagenomes" is ambiguous.

- Response: This is a great question, thank you for raising it. In the updated manuscript, we utilized the same single copy gene normalization analysis to quantify organellar genome copies. It appears that the number of plastid genome copies per haploid genome is ~10, and the number of mitochondrial genome copies per haploid genome could be between 1-3, but both copy numbers are elevated in higher latitudes (>50 °N/S). The results are presented in Extended Data Fig. 5B and mentioned in the main text following the information regarding a lack of metaG representation of mitochondrial genomes: "with about 1-3 mitochondrial genome copies per haploid genome (Extended Data Fig. 5B)".

OK, Very interesting.

- "Since organellar transcripts are not typically analyzed in metaT studies based on polyA-enriched libraries, we also examined if the organelle-mapping reads correlate in NCOG data, where we have complementary polyA and ribosomal RNA-depleted data. For whole genome-mapping read abundance, we saw a significant correlation, with higher abundance recorded for ribo-depleted samples (Spearman's rho for mitochondrial data ranging 0.71-0.79, except the least abundant *P. rex* with 0.51, and p-values always <1⁻²⁰; Spearman's rho for plastid data ranging 0.78-0.82 and p-values always <1⁻⁶⁰; polar genomes were omitted from the analysis). For gene-level mapping abundances, we analyzed *P. globosa* genotype 3 and *P. cordata*. Similarly, most genes were significantly correlated (31 of 38 mitochondrial genes with p-value <1⁻⁴, mean rho 0.42; 200 of 232 plastid genes with p-value <1⁻⁴, mean rho 0.47)."

OK. This is interesting.

- Response: *Phaeocystis* is recognized as its own plankton functional type, separate from other nano- and picoplankton (Buitenhuis et al 2013). Therefore, biomass estimates (MAREDAT) should be comparable with genetic markers from in situ environmental studies identified and annotated as *Phaeocystis* spp. (i.e., they represent the same plankton component). The take-home message from Fig 1A is the substantial presence of *Phaeocystis* in marine environments, regardless of the estimates (note that even mean and median biomass estimates differ quite a bit).

OK

- We updated the text to provide information about this resource in the Figure caption: "Interactive tree available via link in Supplementary Results 6." Additionally, all figure texts were legible when printed.

OK

(Remarks on code availability)

Reviewer #2

(Remarks to the Author)

Thank you for addressing all our comments.

(Remarks on code availability)

Reviewer #3

(Remarks to the Author)

(Remarks on code availability)

Version 2:

Reviewer comments:

Reviewer #1

(Remarks to the Author)

The authors have satisfactorily addressed all of my comments. In my view, they have made significant improvements to the manuscript.

I also appreciate their acknowledgment of the recent publication by Chen et al. (DOI: 10.1016/j.isci.2024.110575), which reports an independent genome assembly and analysis of *Phaeocystis globosa*. For transparency, I would like to clarify that I have no involvement whatsoever with that manuscript.

Chen et al. provides valuable evolutionary insights, notably the identification of two ancient whole-genome duplication events that may have led to expanded gene families associated with key biological functions.

However, the two manuscripts are in fact highly complementary. Füssy et al. presents a broader genomic comparison across 13 strains and incorporates metagenome-assembled genomes (MAGs). Their study goes substantially further in terms of comparative scope, functional biogeography, and integration of environmental data. The functional analyses also differ considerably in content and approach.

Taken together, I consider the Füssy et al. manuscript to be a valuable and original contribution that meaningfully advances the current understanding of *Phaeocystis* diversity and ecology.

(Remarks on code availability)

REVIEWER COMMENTS

Reviewer #1 (Remarks to the Author):

The manuscript titled "Genome-resolved biogeography of Phaeocystales, cosmopolitan bloom-forming microalgae" explores the global distribution and ecological adaptations of the genus *Phaeocystis* through genomic, metagenomic, and transcriptomic analyses and comparisons. The authors utilize large-scale in situ data from expeditions and in vitro culture data from multiple *Phaeocystis* strains. This study is distinguished by its integrative approach.

The authors utilized genomic data from thirteen strains of five *Phaeocystis* species. These data were assembled and annotated, providing a solid foundation for phylogenetic and biogeographic analysis. The analyses also include 21 genomes assembled from metagenomes (MAGs) collected during various expeditions. These MAGs expand the taxonomic sampling and include uncultivated species.

Transcriptomic data were generated under controlled conditions to study the responses of *Phaeocystis* to different levels of nutrients and light. These experimental data provide valuable insights into the regulatory mechanisms and physiological adaptations of *Phaeocystis*, complementing environmental observations.

This manuscript presents an impressive accumulation of analyses and figures; however, for almost all analyses, the conclusions are very general, vague, or the authors fail to reach definitive conclusions. This sentiment is evident from the abstract, where it is clear what the authors have accomplished and the potential value of the new data. However, it is difficult to discern what specific answers the authors provide regarding a better understanding of the biogeochemical cycles of carbon and sulfur.

Response: We appreciate this perspective and agree that at times it might be difficult to easily infer certain bite-sized take-home messages in our manuscript. That being said, upon further reflection and scrutiny we identify a large number of specific answers that advance the way we understand and will ultimately be able to study *Phaeocystis*-driven elemental cycling. The key advancements and conclusions of the study include: a greatly improved genomic resource for Phaeocystales (l.150); phylogenomic evidence and divergence times for Phaeocystales (l.140); species (genotype) level biogeography and niche partitioning (l. 189), also documenting a new cold-water lineage (l. 173); variable mitochondrial-to-plastid expression ratio and its apparent lineage-specific importance (l. 206 and 213); evidence of mixotrophy from transcriptomic comparisons and metabolic models (l. 221 and 224); adaptive gene expression in the global ocean (flavodoxin under iron limitation, metE under B₁₂ limitation, and dsyB/Alma1 for organosulfur production in the polar regions; l. 230), with potential consequences for competition with other algae (l. 238); comprehensive transportome (l. 250), gene family, and horizontal gene transfer analyses (l. 259,

l. 264, l. 267, l. 280 and further; here, linking genes and functions is not possible without detailed knowledge of expression patterns and protein localization, so we must be conservative in our conclusions); *P. globosa* pilot transcriptomics in response to nitrogen input (l. 351) and light regime switch (l. 355), and transcriptomic dynamics (l. 359); and a detailed analysis of the structure (l. 315, l. 330), phylogeny (l. 324, l.331), and expression of endogenous viruses (l. 319, l. 337 on). We acknowledge that these analyses do not always transpire as conclusive findings; however, we believe that the depth and quality of our analyses are unusually robust; many of these topics are expanded in Supplementary Results. Some modifications to the narrative to improve clarity about specific conclusions include the following.

We expanded the conclusion on l. 189:

"Phaeocystales thus emerge as a ubiquitous component of global pico/nanophytoplankton, **with apparent lineage-specific preferences.**"

We newly conclude on l. 227:

"We hypothesize that variable rates of mixotrophy and mitochondrial transcription contribute to this flexibility, and affect ecological niche partitioning between *Phaeocystis* lineages."

Without additional controlled experiments, quantification of the impact on biogeochemical cycles would prove challenging; however, we believe that our genomic and transcriptomic data from a range of laboratory and field studies will indeed inform targeted studies in the future; we pinpointed several directions which such studies could target.

It would undoubtedly be more useful to explore fewer questions but in greater depth. For instance, the analyses on gene function repertoires, gene expansions, and horizontal gene transfer (HGT) are very interesting. Since the authors use transcriptomic data, we expect an analysis of gene expression levels as well as an integration of the physicochemical environmental data. I encourage the authors to better exploit the immense quality of the data to draw more synthetic and clear conclusions on the explanation of biogeography and/or major biogeochemical and/or evolutionary functions.

Response: Inferring differential gene expression in a global metatranscriptomic setting is challenging, particularly because most of the genes are detected in low abundance. The plot below shows a histogram of genes detected by least 10 raw reads across BioSamples (of all 565 sampled), illustrating the low genome coverage by the metaT data:

Typically, genes with such low coverage are removed from differential expression analyses (e.g., edgeR), and therefore could not be subject to more detailed scrutiny in this work. The functions of the top detected genes are summarized in Extended Data Figure 6.

However, we do provide some insights on nitrogen and iron uptake on global scale, namely in relation to overall patterns of mitochondrial transcription (Results I. 211); mixotrophy (Results I. 219-224 and Supplementary Results 5 – Genome-scale metabolic models); ocean-specific adaptations (Results I. 226-232 and Supplementary Results 8 – Physiological responses of Phaeocystales in polar and temperate oceans); and using pilot transcriptomic data from *P. globosa* Pg-G(A) (Supplementary Results 7), where we also described the dynamic transcriptome allocation under specific conditions (Extended Data Fig. 7d,e).

Further, we provide a revised and more in-depth analysis of gene family expansions in the Supplementary Results 6; both species specifically and in the evolutionary context of other algal groups (6 paragraphs/86 lines of analysis). We believe that we have adequately used the potential of the data at hand, while avoiding over-interpretation.

To make the narrative more cohesive, we revisited our analysis of horizontal gene transfers in more detail and added the following to Supplementary Results 6:

“A total of 183 horizontal gene transfer (HGT) events were recorded, passing our most stringent criteria (see Supplementary Methods 8). Most HGTs originated in stramenopiles, dinoflagellates, and opisthokonts. A similar number of events (221), where ingroup monophyly was seen, was classified as acquired ancestrally at the Haptophyta level, with a various number of haptophyte sister sequences (mean=1.73; stdev=3.81). Functionally, the HGT genes contribute to a variety of functions (Fig. 5) without a clear pattern, suggesting a largely stochastic process. Yet, three HGTs might have imparted significant new functions to *Phaeocystis* (Supplementary Table S8). A plastid-targeted thiamine thiazole synthase (THI4) was acquired by *P. antarctica* and *P. globosa*; this

enzyme catalyzes a thiamine biosynthesis (vitamin B1) reaction and has a crucial function for energy metabolism in plants and algae. Consistently, the *Phaeocystis* homologs are relatively well detectable in metaT data (mean RPM 0.44-0.74). Among haptophytes, Pavlovales use a more distantly related THI4, whereas other Prymnesiophyceae employ a different enzyme Thi6, or rely on pyrimidine precursor uptake[REF]. Furthermore, *P. antarctica* acquired a NADP-dependent amino acid dehydrogenase of the Glu/Leu/Phe/Val dehydrogenase family (glutamate according to the best BLAST hit); the gene is duplicated on scaffold 11, and one of the copies appears to have a plastid-targeting presequence. The genes are well detectable in metaT data (mean RPM 0.73/0.74) and may partake in anabolic ammonium assimilation or nitrogen redistribution. *P. globosa* and *P. cordata* only seem to use FAD and NAD-dependent amino acid dehydrogenases. Finally, the most straightforward adaptive novelties of *P. antarctica* are ice-binding proteins (IPR021884; 20 genes), unseen in the warmer-water algal genomes assayed here, yet common in polar algae (<https://itol.embl.de/tree/7844241140296371637243949>). In our data, we detected ice-binding proteins with 0-0.18 mean RPM.”

Additionally, to provide environmental context to Pfam family expansions/contractions, we added mean *in situ* expression information (RPM) to Supplementary Table S8.

The authors analyze and compare relative abundances based on sampling, which results in compositional data. Numerous articles have been published in recent years highlighting the many inherent biases in compositional data, which can lead to erroneous conclusions when compared. There are statistical methods available to correct or mitigate these biases. For example, see doi.org/10.1038/s41467-020-17041-7. Therefore, I wonder if, for some analyses at least, validation of the absence of bias would be necessary.

Response: We appreciate the raised concern and acknowledge that statistical analyses of compositional data are prone to biases, particularly in the absence of quantitative standards. For the metatranscriptomic and metagenomic analyses we used TPM normalization, which is commonly used when quantitative standards are not available (Cohen et al., 2022; doi: 10.3389/fmars.2022.867007).

For the *Phaeocystis globosa* transcriptomic analyses, we also used TPM normalization, because it is suitable for variable transcript allocation analysis (Alexander et al., 2015; doi: 10.1073/pnas.1518165112). We also evaluated our data using TMM transformation of the edgeR package, which is well-suited for transcriptomic analyses with varying biological replicates, and obtained similar expression patterns for biological processes, except for the single cell / colony comparison, where 99% of the DE ortholog expression showed downregulation in the TMM data, and did not seem to support overall biological expectations from this morphotype switch. When correlating expression on the pathway level (i.e. covariation of genes that belong to pathways mapping at least 0.5% of the total expression; 32 pathways in the TPM matrix and 30 pathways in the TMM matrix), TPM showed higher levels of correlation than the TMM-normalized

matrix, indicating that the underlying gene expression was more homogeneous under TPM normalization:

Many figures are much too small.

Response: We adjusted the figures and believe they comply with the Nature Publishing Groups guides for preparing figures, including font size (5-7pt). Some figures with multiple panels were adapted from published work (e.g., DOI: 10.1038/s41467-017-02342-1, Fig.6), and the sizes of the panels are comparable. Furthermore, we verified text legibility in print. We are happy to discuss the matter more specifically with the copy editor in due course and/or to respond to any specific suggestions that arise during the review process.

The various cultures sequenced in this study were done so using very different sequencing and assembly protocols. Although I do not think this affects the main conclusions, these technical differences can lead to varying quality levels across genomes and, consequently, different biases, particularly in repetitive regions, gene family expansions, etc. The authors should comment on this point.

Response: We tried to estimate the repetitive content in the cultured strains using various tools (Supplementary Table S1-S2) and have further updated this with k-mer abundance analyses using raw reads (GenomeScope2). The assembly-independent analyses indicate that there is some variability in (haploid) genome size and simple repeat content among the lineages (*P. globosa* genotype 1 – 6 different strains, typically between 86.3-185.0 Mbp, 38-69% repetitive content for the best-model-fit estimates; *P. globosa* genotype 2 98.6-107.6 Mbp, 37% repetitive; *P. globosa* genotype 3 92.9-103.4, 40% repetitive; *P. antarctica* – with a poor model fit – 142.6-147.2 Mbp, 39% repetitive; *P. rex* 190.9-237.8 Mbp, 56% repetitive; *P. cordata* 71.8-75.8 Mbp, 14% repetitive; *P. jahnii* 136.9-146.4, 40% repetitive). More detailed results are below:

Code	Species	K-mers	Homozygous [%]	Heterozygous [%]	Genome Haploid Length [bp]	Genome Repeat Length [bp]	Rep [%]	Genome Unique Length [bp]	Model Fit [%]	Read Eri Assembly [bp]	
GBNOY	P.globosa genotype 1	21	97.2417 - 97.8493	2.15067 - 2.75828	85,580,786 - 93,068,397	31,907,031 - 34,698,632	37.28	53,673,755 - 58,369,765	2.6487 - 3.20256	0.63%	141,838,632
GBNOY	P.globosa genotype 1	23	99.0051 - 99.6459	0.354052 - 0.994937	154,839,101 - 171,599,373	107,478,726 - 119,035,685	69.37	47,460,375 - 52,563,688	0.729263 - 0.148907	0.47%	141,838,632
GBNOY	P.globosa genotype 1	25	98.9938 - 99.5731	0.426903 - 1.00619	155,386,906 - 172,008,655	106,587,603 - 117,989,287	68.59	48,799,303 - 54,019,368	0.084537 - 0.164071	0.48%	141,838,632
GBNOZ	P.globosa genotype 1	21	96.3219 - 100	0 - 3.67806	86,403,634 - inf	51,465,413 - inf	59.56	34,938,221 - inf	3.18032 - 3.60498	0.45%	139,323,495
GBNOZ	P.globosa genotype 1	23	98.2689 - 99.6316	0.368369 - 1.73108	130,680,974 - 185,082,087	86,892,414 - 123,064,811	66.49	43,788,560 - 62,017,276	0.990457 - 0.33508	0.47%	139,323,495
GBNOZ	P.globosa genotype 1	25	91.8165 - 100	0 - 8.18349	52,371,998 - inf	26,323,343 - inf	50.26	26,048,655 - inf	3.01158 - 5.82816	0.48%	139,323,495
GBNPA	P.globosa genotype 3	21	98.3964 - 98.9679	1.03211 - 1.60358	92,918,466 - 100,590,424	37,882,376 - 41,010,843	40.77	55,035,490 - 59,579,581	2.30038 - 2.92416	0.43%	111,385,349
GBNPA	P.globosa genotype 3	23	98.4576 - 99.0028	0.997213 - 1.5424	94,168,237 - 102,545,456	38,447,344 - 41,867,625	40.83	55,720,893 - 60,677,831	2.17401 - 2.57067	0.44%	111,385,349
GBNPA	P.globosa genotype 3	25	98.4758 - 99.0235	0.97649 - 1.52418	94,217,828 - 103,400,735	37,348,881 - 40,989,077	39.64	56,868,946 - 62,411,658	3.12183 - 1.0771	0.44%	111,385,349
GBNPB	P.rex	21	97.7204 - 98.7378	1.26217 - 2.27962	196,052,837 - 233,783,332	112,194,429 - 133,786,320	57.23	83,858,408 - 99,997,013	1.05148 - 1.17816	0.71%	179,384,885
GBNPB	P.rex	23	97.7467 - 98.7735	1.22647 - 2.25331	193,585,728 - 234,845,530	108,116,901 - 131,160,345	55.85	85,468,827 - 103,685,185	0.45351 - 0.12217	0.74%	179,384,885
GBNPB	P.rex	25	97.7747 - 98.8355	1.16449 - 2.22528	190,932,233 - 237,804,729	105,414,850 - 131,293,441	55.21	85,517,383 - 106,511,288	0.37006 - 0.391285	0.77%	179,384,885
GBNPH	P.globosa genotype 1	21	97.7359 - 100	0 - 2.2641	99,894,582 - 400,087,888	47,594,696 - 190,621,564	47.64	52,299,886 - 209,466,324	4.17517 - 0.887995	0.47%	143,081,016
GBNPH	P.globosa genotype 1	23	96.3274 - 100	0 - 3.67255	84,382,866 - 5,920,820,288	40,297,307 - 2,827,506,632	47.76	44,085,559 - 3,093,313,656	3.74958 - 1.31307	0.48%	143,081,016
GBNPH	P.globosa genotype 1	25	96.615 - 100	0 - 3.38501	85,311,038 - 31,806,031,881	38,836,072 - 14,479,033,202	45.52	46,474,966 - 1,326,998,679	3.17041 - 1.77251	0.49%	143,081,016
GBNPN	P.globosa genotype 2	21	98.862 - 99.4126	0.58742 - 1.13796	98,592,706 - 105,015,820	36,546,701 - 38,927,645	37.07	62,004,004 - 66,088,176	2.24781 - 2.95892	0.44%	105,649,752
GBNPN	P.globosa genotype 2	23	98.8911 - 99.4164	0.583569 - 1.10893	99,615,054 - 106,509,411	36,345,830 - 38,861,325	36.49	63,262,223 - 67,648,086	1.78832 - 2.25706	0.45%	105,649,752
GBNPN	P.globosa genotype 2	25	98.9248 - 99.4209	0.57908 - 1.07542	100,325,837 - 107,625,523	36,543,256 - 39,202,125	36.42	63,782,207 - 68,423,397	1.67776 - 2.08109	0.54%	105,649,752
GBNPP	P.globosa genotype 1	21	97.244 - 100	0 - 2.75602	98,266,716 - 573,818,060	56,968,311 - 332,680,408	57.97	41,298,405 - 241,157,852	2.53862 - 1.87352	0.46%	137,791,476
GBNPP	P.globosa genotype 1	23	97.0457 - 100	0 - 2.95426	94,293,474 - 1,092,925,371	53,669,830 - 622,069,762	56.92	40,623,644 - 470,855,609	1.40726 - 5.13978	0.47%	137,791,476
GBNPP	P.globosa genotype 1	25	95.6553 - 100	0 - 4.34471	75,949,026 - inf	37,735,399 - inf	49.32	38,493,964 - inf	2.44621 - 2.03368	0.48%	137,791,476
GBNPS	P.globosa genotype 1	21	95.3194 - 100	0 - 4.6806	72,549,529 - inf	37,735,399 - inf	52.01	34,814,131 - inf	4.16755 - 1.87832	0.42%	128,432,992
GBNPS	P.globosa genotype 1	23	95.4557 - 100	0 - 4.5443	71,763,956 - inf	36,695,622 - inf	51.13	35,068,335 - inf	3.14561 - 2.34838	0.43%	128,432,992
GBNPS	P.globosa genotype 1	25	98.1775 - 99.4216	0.578444 - 1.82254	116,845,611 - 164,968,169	75,471,633 - 104,761,270	63.5	43,373,978 - 60,206,899	1.53738 - 0.214204	0.44%	128,432,992
GBNPT	P.globosa genotype 1	21	97.2746 - 98.1112	1.88883 - 2.72537	86,640,302 - 98,460,163	33,750,576 - 38,354,982	38.95	52,888,726 - 60,105,181	0.257908 - 0.0449764	0.58%	140,731,861
GBNPT	P.globosa genotype 1	23	97.2021 - 98.2944	1.70565 - 2.79786	86,260,153 - 103,201,694	32,624,157 - 39,031,559	37.82	53,635,996 - 64,170,135	0.0658597 - 0.0755298	0.48%	140,731,861
GBNPT	P.globosa genotype 1	25	95.3345 - 99.9506	0.0494058 - 4.66552	71,396,534 - 146,411,648	18,822,620 - 38,599,220	26.36	52,570,923 - 107,812,428	0.31477 - 0.108676	0.48%	140,731,861
GCBAO	P.jahniil	21	98.1443 - 98.5772	1.42799 - 1.85574	136,912,341 - 145,341,382	56,553,809 - 60,035,557	41.31	80,358,533 - 85,305,825	3.17106 - 4.06165	0.40%	160,177,556
GCBAO	P.jahniil	23	98.195 - 98.6023	1.39774 - 1.805	137,440,173 - 146,039,300	54,396,391 - 57,799,773	39.58	83,043,782 - 88,239,526	0.75724 - 0.679527	0.42%	160,177,556
GCBAO	P.jahniil	25	98.2625 - 98.6545	1.34554 - 1.73755	137,242,443 - 146,384,021	53,461,413 - 57,022,423	38.95	83,781,029 - 89,361,598	0.434097 - 0.330138	0.43%	160,177,556
Phagio	P.globosa genotype 3	21	78.8068 - 100	0 - 2.11932	58,444,604 - 73,522,566	27,248,174 - 34,277,855	46.62	31,196,430 - 39,244,710	75.2876 - 96.2571	6.66%	155,975,881
Phagio	P.globosa genotype 3	23	69.9356 - 100	0 - 30.0644	58,299,398 - 79,630,359	26,876,093 - 36,709,691	46.1	31,423,306 - 42,920,668	46.1 - 96.266	5.80%	155,975,881
Phagio	P.globosa genotype 3	25	98.9022 - 100	0 - 1.0978	94,910,876 - 149,669,493	27,213,257 - 42,913,885	28.67	67,697,619 - 106,755,609	74.3256 - 95.7406	3.54%	155,975,881
Phacord	P.cordata	21	98.9743 - 99.4662	0.533782 - 1.02572	71,894,290 - 75,659,231	10,711,380 - 11,272,324	14.9	61,182,825 - 64,386,907	3.63614 - 3.91512	0.37%	86,876,152
Phacord	P.cordata	23	99.0336 - 99.5017	0.498271 - 0.966389	71,751,490 - 75,741,835	9,742,198 - 10,283,994	13.58	62,009,226 - 65,457,841	3.33801 - 3.57111	0.48%	86,876,152
Phacord	P.cordata	25	99.0748 - 99.5044	0.495637 - 0.925228	71,788,765 - 75,811,915	9,531,265 - 10,065,411	13.28	62,257,500 - 65,746,504	2.54782 - 2.72188	0.46%	86,876,152
Phaant	P.antarctica	21	99.1161 - 99.1864	0.813647 - 0.883908	142,686,567 - 143,884,924	57,178,640 - 57,658,857	40.07	85,507,927 - 86,226,067	66.7358 - 95.6466	3.00%	199,073,031
Phaant	P.antarctica	23	99.1443 - 99.2102	0.789788 - 0.8557	144,556,020 - 145,824,977	56,686,206 - 57,183,815	39.21	87,869,814 - 88,641,162	67.5184 - 95.8052	3.02%	199,073,031
Phaant	P.antarctica	25	99.1715 - 99.2341	0.765853 - 0.82849	145,928,967 - 147,273,921	56,228,538 - 56,746,768	38.53	89,700,430 - 90,527,153	68.2242 - 95.9479	3.02%	199,073,031

Note that these estimates seem affected by read mapping error rates (Phagio1 and Phaant1) and do not always fit the general model function of GenomeScope2 the same way for the individual k-mer analyses (21,23, and 25; *P. globosa* genotype 1), which complicates the estimation.

We stated that the high simple repeat content in *P. globosa* strains complicated their assembly. This was the main reason why we did not attempt gene annotation and subsequent gene family analyses in these cases. We briefly mention these obstacles in the Results of the original manuscript (currently on I.118):

“Specifically, repetitive elements make up 35% (55 Mbp) of Phagio1 and 50% (101 Mbp) of Phaant1, which partially explains the higher genome size of Phaant1, and the fragmentation of assemblies when using short reads only (see Methods).”

We added the following narrative to Supplementary Results 1 to reflect on these problems:

“... likely stems from the limitations of short read-only data assembly. Indeed, according to GenomeScope 2.0 estimates, substantial portions of the data represent short repeats (37-69%; not shown). Due to these technical problems and limitations, repetitive regions remain difficult to assemble and quantify in *Phaeocystis* spp., along with recently duplicated genes they might flank. The varying levels of assembly quality introduce biases in coding and non-coding repeats, which is why we did not attempt to quantify gene family expansion in the fragmented assemblies.”

The gene prediction method is a crucial and very delicate step for environmental eukaryotic genomes. The authors simply cite "phytozome and mycozome pipelines." References 9 and 10 indicated are not methodological pipelines; they are genome databases. The authors must clearly explain how the exon-intron structures of nuclear genes are predicted. Are introns present (the word is completely absent from the main and supplementary text)? Additionally, I do not see any statistics on gene and protein sizes anywhere.

Response: The two gene prediction pipelines referenced are underlying the databases, although were not specifically described in the articles. These are, however, the preferred references for the authors (from the Joint Genome Institute – JGI) of the pipeline who are co-authoring this manuscript. We added the following information to Supplementary Methods:

“The two annotation pipelines employ different intron-aware gene-finding worker algorithms, but fundamentally use a combination of homology-based modeling, *ab initio* predictions, and RNA/transcriptome evidence methods. Filtering combines different support metrics to select an appropriate single model to represent each locus. “

Gene structure statistics for the three reference genomes and Emihu1 for comparison are now tabulated in Supplementary Table S1.

The analysis of viral sequence integrations does not incorporate a significant collection from the new clade of mirusvirus (doi: 10.1038/s41586-023-05962-4), nor the following collection for NCLDV: doi.org/10.1038/s41586-020-2924-2.

Response: As for the first reference, the dataset does not contain Yaravirus-like or Pleurochrysis sp. endemic viruses 1a and 2, since, as we elaborate in Supplementary Results 2., the key phylogenetic markers used in the study (DNAPolB, RNAPa, RNAPb, TFIIS) are missing in Phaeocystis endogenous NCLDVs and other Yaravirus-like viruses. The study therefore does not provide a suitable phylogenetic context for our endogenous NCLDVs (the closest relatives from the dataset would be Asfaviridae with a relatively long branch). The second reference focuses on Mimiviridae, endogenous NCLDVs only distantly related to the Phaeocystis NCLDVs described in our manuscript. Nevertheless, we updated the phylogeny in Extended Data Fig S8B and added references to these recent works to account for the development in the NCLDV field.

The in silico metabolic reconstruction is very interesting. Can we associate these differences in carbon and nitrogen sources with environmental parameters and evolutionary proximities between lineages or species?

Response: We developed these preliminary models to address specific questions, i.e., to predict the fluxes/relative fluxes in different conditions and for estimating mitochondrial/plastid flux ratios. These questions did not necessitate manual curation of the gene-reaction association, which is a time-consuming process, but as a result they do not allow linking fluxes with any gene-related data (e.g. transcriptomic data) or for finding marker genes for a trophic mode of these species, and can lead to erroneous outcomes. Based on our experience with *Cylindrotheca* (Kumar et al., 2024; doi: 10.1126/sciadv.ado2623), curating these models could take up to 3-4 months, with an additional 3-4 weeks to analyze the model predictions and validate them with environmental data. While these results are surely interesting, we would prefer to address them in a future work; perhaps with a specific focus on mixotrophy in nanoflagellates.

As for evolutionary proximities, *P. antarctica* and *P. globosa* are more closely related and diverged more recently (see Figure 1), but *P. globosa* inhabits environments more similar to *P. cordata* and other *Phaeocystis*. The differential flux response to organic carbon and nitrogen (Extended Data Fig. 7b) might be driven by specialization to temperate vs. polar regions, whereas other fluxes through nitrogen metabolism might reflect evolutionary distance (Extended Data Fig. 7c).

We added the following sentence to Supplementary Results 5:

“Our *Phaeocystis* GEMs suggest variations in metabolic arrangements that might be driven by both environmental adaptation (e.g., the effect of organic nitrogen to M/P ratio) and evolutionary distance (e.g., fluxes through nitrogen metabolism).”

Details:

Abstract:

"highest-contiguity" is relative and subjective. It's better to use more precise terms.

"fine-tuned preferences" – Use another term than fine-tuned, as it can sometimes be misinterpreted (e.g., intelligent design).

We replaced “three [genomes] with the highest contiguity” with “three with N50 >30,000”, and replaced “fine-tuned” with “lineage-specific”.

Text:

"The gene repertoires of three annotated *Phaeocystis* reference genomes reflect a unique strategy of gene expansion among algae: repetitive elements, horizontal gene transfer, and full-length endogenous virus insertions." Do we have enough sequenced algae (prokaryotes and eukaryotes) to support this statement?

Response: Rephrased to a non-comparative statement:

“The gene repertoires of three annotated *Phaeocystis* reference genomes reflect an adaptation strategy through gene expansion: repetitive elements, horizontal gene transfer, and full-length endogenous virus insertions.”

The statement "Phaeocystales thus emerge as an ecologically versatile group with diverse adaptations to biotic and abiotic stressors" is very vague and general. It is hard to imagine many groups for which these conclusions would not be true.

Response: That is probably true, but many of the phenomena we analyze (mixotrophy, endogenous viruses, functional gene family expansion) are emergent topics in algal genomics, and we anticipate their importance will grow with more experimental data. This sentence merely concludes the abstract, and we see it as a teaser to raise interest in the rest of the article.

Line 153: The placement of references 46 and 47 in the sentence creates ambiguity about their role. I suggest adding citations of examples from the

numerous references in the literature on the biogeographic distribution of eukaryotic phytoplankton based on barcodes in the first part of the sentence. Furthermore, biogeographic data on Phaeocystis from genome data, particularly MAGs, already exists. Reference 29, among others, provides numerous maps (or links towards maps) of this lineage and should therefore be discussed in the introduction at this point in the manuscript.

Response: Thank you for this remark, we amended the references as suggested. While Delmont et al. 2022 provide biogeography data for their metagenome-assembled genomes (MAGs), Phaeocystales MAGs were relatively incomplete (see Supplementary Table S1, sheet MAGs), and were not analyzed in more detail. We made a reference to this work in the Introduction: “While reports on their global biogeography exist (REFs), they are based on amplicons or partial metagenomes, and do not elaborate on gene-level adaptation.”

Lines 203 and the subsequent discussion are very interesting. Have the authors analyzed the ratio of genomic abundance between the nucleus and organelles? This analysis could reveal variations in the number of organelles per cell, providing significant insights into the transcriptomic analysis and potentially highlighting important aspects of cellular regulation and adaptation.

Response: This is a great question, thank you for raising it. In the updated manuscript, we utilized the same single copy gene normalization analysis to quantify organellar genome copies. It appears that the number of plastid genome copies per haploid genome is ~10, and the number of mitochondrial genome copies per haploid genome could be between 1-3, but both copy numbers are elevated in higher latitudes (>50 °N/S). The results are presented in Extended Data Fig. 5B and mentioned in the main text following the information regarding a lack of metaG representation of mitochondrial genomes: “with about 1-3 mitochondrial genome copies per haploid genome (Extended Data Fig. 5B)”.

We also examined if organelle-mapping metaT reads correlated between polyA-enriched and ribosomal RNA-depleted samples. We added the following to Supplementary Results 3:

“Since organellar transcripts are not typically analyzed in metaT studies based on polyA-enriched libraries, we also examined if the organelle-mapping reads correlate in NCOG data, where we have complementary polyA and ribosomal RNA-depleted data. For whole genome-mapping read abundance, we saw a significant correlation, with higher abundance recorded for ribo-depleted samples (Spearman’s *rho* for mitochondrial data ranging 0.71-0.79, except the least abundant *P. rex* with 0.51, and p-values always <1⁻²⁰; Spearman’s *rho* for plastid data ranging 0.78-0.82 and p-values always <1⁻⁶⁰; polar genomes were omitted from the analysis). For gene-level mapping abundances, we analyzed *P. globosa* genotype 3 and *P. cordata*. Similarly, most genes were significantly correlated (31 of 38 mitochondrial genes with p-value <1⁻⁴, mean *rho* 0.42; 200 of 232 plastid genes with p-value <1⁻⁴, mean *rho* 0.47).”

Figure 1A: The bar plots for the unigene collection represent the composition of the collections, not the communities. To enable a useful comparison between the 'maredat' and 'unigene collections' bar plots, the figures should represent the same type of information.

Response: *Phaeocystis* is recognized as its own plankton functional type, separate from other nano- and picoplankton (Buitenhuis et al 2013). Therefore, biomass estimates (MAREDAT) should be comparable with genetic markers from *in situ* environmental studies identified and annotated as *Phaeocystis* spp. (i.e., they represent the same plankton component). The take-home message from Fig 1A is the substantial presence of *Phaeocystis* in marine environments, regardless of the estimates (note that even mean and median biomass estimates differ quite a bit).

Additionally, the violin plots in Figures 1A and 1B display negative values, which is clearly an artifact of the representation. This issue must be corrected to ensure the accuracy and clarity of the visual data presentation.

Response: Fixed.

Figure 1h: A significant zoom is required to discern the information in this figure, but at this level of magnification, the image becomes blurry. Improving the resolution or providing a more focused view would enhance the clarity and usefulness of this figure.

Response: All figures are provided as vector images, and the fonts are consistent with the requirements (5-7pt). There should be no blur if given enough time to render. We will also discuss this matter with the copy editor in due course if provided the opportunity. There is no panel h in Figure 1, so we are not certain about which specific information this concerns. If the Reviewer meant Figure 5h, there is an interactive tree referenced at the end of Supplementary Results 6 – Gene family expansion. We updated the text to provide information about this resource in the Figure caption: “**Interactive tree available via link in Supplementary Results 6.**” Additionally, all figure texts were legible when printed.

Figure 5 Legend: It is unclear how the dataset could originate from reference 56 as indicated. Please verify the source of the dataset and correct the reference if necessary.

Response: Should be ref⁶⁷ (Dorrell et al., 2023). Thank you for spotting this broken reference link.

More methodological details on the analysis of environmental drivers using Generalized Additive Models (GAM) need to be provided. This should include information on the specific variables considered, the model fitting process, and any validation or testing procedures employed to ensure the robustness of the analysis

Response: The main reason for utilizing GAMs was that we noticed modal distribution of *Phaeocystis* abundance across multiple independent environmental variables, and therefore needed non-linear statistics. The abundance data for each lineage were tested for normality prior to and after logarithmic and box-cox transformation. While normality was rejected by Shapiro-Wilk tests in most cases, the skewness and kurtosis of the transformed data did not differ substantially from normal distribution (i.e., were around 0 and 3, respectively, see table below). The transformed data did not fit to gamma distribution either. The environmental parameters used were not multi-collinear.

	P.glo1	P.glo2	P.glo3	P.antarctica	P.poucheiii	P.rex	P.cordata	P.jahnii	PSC
skewness									
original	1.93	1.53	2.84	4.64	2.90	4.06	1.76	8.54	2.56
log	-0.33	-0.17	0.52	1.01	0.82	0.56	-0.40	-0.05	-0.64
box-cox	0.03	0.16	1.11	1.85	1.09	1.15	0.11	1.26	0.09
residuals (log-trans)	-0.27	-0.34	-0.29	-0.54	-0.54	0.35	-0.23	-0.12	-0.51
kurtosis									
original	8.43	5.00	10.45	25.90	12.34	22.49	5.88	79.38	9.66
log	2.07	1.71	3.25	3.47	1.99	3.22	2.54	3.51	2.67
box-cox	2.08	1.71	4.17	6.34	2.60	4.71	2.30	7.45	2.47
residuals (log-trans)	2.75	3.21	3.47	3.74	3.45	4.06	3.59	2.55	4.03
AIC - GAM									
original gauss	1147.6	1429.1	1136.0	1580.9	1704.9	1278.9	1373.0	1329.7	1230.0
original gamma	993.4	1247.7	944.3	1052.0	909.4	1073.7	1223.1	983.7	958.4
log gauss	219.7	289.0	211.7	243.3	253.1	246.6	288.4	303.7	294.8
log gamma	226.3	293.8	213.5	253.5	294.2	234.4	288.3	333.1	284.2
box-cox gauss	383.2	478.1	369.6	405.2	452.0	420.6	483.1	445.9	445.8
box-cox gamma	372.7	484.3	355.3	406.7	396.5	399.0	469.7	451.0	393.5

Finally, we decided to log-transform the abundance data, considering the AIC values suggested a better fit for this input. Consistently, the residuals had a near-normal distribution as well (based on skewness and kurtosis).

Added to Methods: “The normalized read count data had an approximately normal distribution after log transformation (based on skewness ~0 and kurtosis ~3).” and “The fitting parameters included: $\text{gam}(\text{logtransformed_normreads} \sim \text{s}(\text{logChl_a}) + \text{s}(\text{Iron_5m}, k = 5) + \text{s}(\text{Nitrate_5m}, k = 5) + \text{s}(\text{log}(\text{Ammonium_5m})) + \text{s}(\text{diatoms}, k = 5) + \text{s}(\text{dinophyceae}, k = 5) + \text{s}(\text{Temperature}) + \text{s}(\text{Distance_coast}))$. Statistics were also calculated individually for each independent variable.”

Reviewer #2 (Remarks to the Author):

This is a solid seminal work bringing together a wealth of genomic and metagenomic information available on *Phaeocystis*. The authors review and refine the phylogeny and biogeography of this globally significant genus of phytoplankton.

The authors conducted a detailed exploration of the global biogeography and adaptive strategies of *Phaeocystis* to variations in nutrient availability and form, temperature and environmental preferences through the genomes of several significant cultured species of *Phaeocystis*. They did this in combination with MAGs from several global expeditions to accommodate uncultured species. The data also provided cues on the varying levels of heterotrophy, metabolic flexibility and trophic strategies between the species/strains. Additionally, the authors also compared viral footprint integration between the *Phaeocystis* species and explored the interactions between virus and host through transcriptomic data.

We found the study to be novel, comprehensive and informative, contributing a significant amount of new data and analyses towards a better understanding of *Phaeocystis*. The study investigated the adaptive mechanisms and strategies of *Phaeocystis*, which will be key to understanding and predicting future shifts in *Phaeocystis* distribution and their contribution to global nutrient cycling, carbon sequestration, etc. We found the synthesis of results and findings around mixotrophy and strategies to circumvent iron and vitamin B12 limitations particularly exciting.

Response: Thank you for your positive feedback, we are also excited about the new opportunities *Phaeocystis* research might offer.

The MS is clear, logical, easy to follow. We found it was very heavy on acronyms though which made it occasionally hard to follow. We suggest that the authors compile a list of acronyms used in the MS and their definitions on a single page that can be referred to on the first page of supplementary methods, for easier reading.

Response: We have added this information to the beginning of the Supplementary Information.

Specific comments:

Title/abstract:

It might help the reader if the authors explained upfront that Phaeocystales consisted of a single known genus plus a distantly related environmental sister clade (which would be a separate genus, based on genetic distance). At first we found it confusing that the title mentioned an order, but all the results seemed to talk of a single genus.

Response: Thank you for pointing this out. We now open the abstract with the following information to clarify this point.

“Phaeocystales, comprising a single genus *Phaeocystis* and an undescribed sister lineage, are nanoplanktonic haptophytes...”

L107: We could not find MMETSP defined anywhere in the main text or supplementary materials.

Response: Fixed.

L151: The two sentences seem to be contradictory to each other?

Response: The reviewer refers to “Biogeographic studies of eukaryotes traditionally rely on **sequencing** short regions of universal marker genes **via metabarcoding** that cannot fully resolve phytoplankton diversity or capture physiological responses. Consequently, there has been an unprecedented accumulation of metatranscriptomic (metaT) and metagenomic (metaG) data from various environments that allow for higher taxonomic and functional resolution and *in situ* physiological responses of whole communities”. We added the highlighted words and hope the new formulation is clearer; we are trying to say that amplicon sequencing does not always provide sufficient species resolution or clues for community functions and physiology, and that meta-omics are increasingly more frequently used to fill these gaps.

L255: We could not find dinoflagellates indicated in the text in Fig. 5. We suppose dinoflagellates are categorized as alveolates, but other subdivisions of alveolates also exist. If only dinoflagellates are found or classified as alveolates in this case, then it should be indicated explicitly in the diagram.

Response: The reviewer is correct that there are other alveolates in the reference database, but in our HGT trees Ciliophora and Apicomplexa occur only in negligible counts, with Dinoflagellata represented in 98.2% of HGT clades in question by at least three species (ciliates would often occur as a result of reference transcriptome contamination). We apologize for this confusion and clarify this point in the revised manuscript in the legend of Figure 5:

“Note that the vast majority (98.2%) of Alveolata sequences detected in the HGT clades belonged to Dinoflagellata.”

Supplementary Results (Endogenous viruses section): PgV was defined, but what is PgVV (also in Extended Data Fig. 8)?

Response: Fixed.

Reviewer #3 (Remarks to the Author):

"I co-reviewed this manuscript with one of the reviewers who provided the listed reports. This is part of the Nature Communications initiative to facilitate training in peer review and to provide appropriate recognition for Early Career Researchers who co-review manuscripts."

Response: Thank you for your time, we hope you found this manuscript a positive review experience.

REVIEWER COMMENTS

Reviewer #1 (Remarks to the Author):

I thank the authors for seriously considering the remarks and questions provided.

Overall, I remain of the opinion that this manuscript presents significant data and an impressive effort involving a large number of analyses.

However, the text often remains rather superficial and does not sufficiently highlight the quality of the data and analyses. A clear message does not emerge, which is exemplified by the abstract and conclusion, where the text remains fairly general.

Response: Thank you for your encouraging feedback. We acknowledge the challenge of distilling the essence of our comprehensive analyses for a broad readership, such as that of *Nature Communications*, within the 150-word abstract limit. However, we have provided a more detailed discussion in the Supplementary Results. In response to your comment, we have refined the main text in several places to minimize remaining generalizations and present concrete conclusions.

The abstract now reads:

Phaeocystales, comprising the genus *Phaeocystis* and an uncharacterized sister lineage, are nanoplanktonic haptophytes widespread in the global ocean. Several species form mucilaginous colonies and influence key biogeochemical cycles, yet their underlying diversity and ecological strategies remain underexplored. Here, we present new genomic data from 13 strains, including three high-quality reference genomes (N50 >30 kbp), and integrate previous metagenome-assembled genomes to resolve a robust phylogeny. Divergence timing of *P. antarctica* aligns with Miocene cooling and Southern Ocean isolation. Genomic traits reveal metabolic flexibility, including mixotrophic nitrogen acquisition in temperate waters and gene expansions linked to polar nutrient adaptation. Concordantly, transcriptomic comparisons between temperate and polar *Phaeocystis* suggest Southern Ocean populations experience iron and B₁₂ limitation. We also identify signatures of horizontal gene transfer and endogenous giant virus/viropage insertions. Together, these findings highlight Phaeocystales as an ecologically versatile and geographically widespread lineage shaped by evolutionary innovation and adaptation to contrasting environmental stressors.

Additional changes were made on l. 192

While Phaeocystales are widely recognized as ubiquitous nanophytoplankton, our findings reveal overlooked lineages with varying abundances, specialized to different environments, and shaped by complex evolutionary trajectories.

On l. 304 we established new connections between gene family expansions in *P. antarctica* and an over-representation of some of these domains in the South Atlantic and Southern Ocean:

The over-representation of additional iron-responsive and organosulfur metabolism genes compared to warmer latitudes (Fig. 3i) suggests that Southern Ocean-specific expansions may underlie the observed higher expression levels, possibly an adaptation to chronic nutrient depletion. In the remaining section about gene/domain family expansions, which we believe is comprehensive and robust, it is worth noting that we do mention specific examples how these expansions might be linked to various traits (both in Results and more extensively in Supplementary Results). For example, we mention “von Willebrand proteins ... found to be iron-responsive and hypothesized to participate in colonial matrix formation”, “nitrite/sulfite reductase and carbonic anhydrase domains ... perhaps enhancing the assimilation capabilities of inorganic nitrogen, sulfur, and carbon”, “ice-binding proteins ... likely crucial for Southern Ocean colonization”, make a connection between “the presence of signal peptides in ShKT-domain proteins [suggesting] a localization via secretory pathways, and iron-dependent expression patterns...[ref125 SI]”, or point out “zinc-finger domains ... enriched in all three genomes”. Furthermore, we picked three examples of *P. antarctica*-specific protein family expansions (nitrite/sulfite reductases, two families of carbonic anhydrases, and vacuolar ion transporter family VIT/Ccc1) and assessed their gene copy numbers in metagenomic data; we observed more gene copies of these families specifically in the Southern Ocean, partly validating our domain enrichment analyses and suggesting their general importance in this environment. We updated Fig. 3i to include this information:

Figure 3: i) Gene expansion of selected gene families as a function of latitude. MetaG reads mapping to these genes were normalized to length and single-copy gene loci. CA, carbonic anhydrase; Nitrite red., nitrite-sulfite reductase; VIT, vacuolar ion transporters.

Additionally, certain details in the results have been modified without explanation. While I understand that these could be drafting errors, the authors should provide justification for these changes, as they sometimes affect the results.

For example:

- Supp Line 601: Ammonium and nitrate have been reversed.
- Supp Line 601: "Dinoflagellate" has been removed.
- Supp Line 615: "Silicate" has been removed.

Response: These changes emerged from data transformation and reanalysis, as requested in the first round of revisions. We apologize for not commenting on these changes, which were no longer statistically significant. We also removed the results regarding abundance vs. tubulin expression, as these were difficult to interpret. Other major insights we had made held following this statistical reassessment.

Some additional details:

- Supp (track change), line 881 : “Notably, however, flavodoxin was significantly downregulated in iron-replete conditions, in line with other results” : Add reference
- Abstract : “N50 >30,000.” : missing unit

Response: added reference to Fig. 3g-i, Supplementary Results S8, and corrected the missing N50 unit here and on I.111 and I.116-117.

And my comments on your responses:

- “We expanded the conclusion on I. 189: Phaeocystales thus emerge as a ubiquitous component of global pico/nanophytoplankton, with apparent lineage-specific preferences.”

The sentence remains very general; it is already well known that Phaeocystales is ubiquitous. Adding 'with apparent lineage-specific preferences' does not provide any additional insight. If different lineages exist, it is self-evident that they would exhibit differences; this is tautological.

Response: Rephrased (mentioned above). Note that while it might be evident that different lineages would exhibit differences, *Phaeocystis* species/lineages are challenging to classify across various environments due to their inconspicuous appearance as nanoflagellates, which apparently makes them difficult to distinguish and study in natural ecosystems.

We added narrative in Results and Supplementary Results 3 to expand on these thoughts:

[Results] Notably, four MAGs not affiliated with the *antarctica/pouchetii* polar clade appear to have a substantial polar presence, suggesting convergent colonization of cold waters (Supplementary Results 3).

...

While Phaeocystales are widely recognized as ubiquitous nanophytoplankton, our findings reveal overlooked lineages with varying abundances and environmental specializations, shaped by complex evolutionary histories. Given that *Phaeocystis* is currently treated as a single PFT in global biogeochemical models, such hidden diversity could have important implications. If members of the clade differ in ecological roles and functional traits, model predictions may be affected. Our results emphasize the value of incorporating multiple, data-informed groups into future *Phaeocystis* experimental frameworks — paralleling efforts to refine models through strain-specific thermal niches of *E. huxleyi* (REF).

[Supplementary Results 3] To assess whether any of the assemblies represent polar specialists, we tested for enrichment of normalized reads from the Southern Ocean (latitude <50 °S) or the Arctic Ocean (latitude >60 °N) relative to non-polar stations (temperate + tropical). Indeed, *P. antarctica* was >500× more abundant in the Southern Ocean (i.e., TARA_SOC_28_MAG_00057: 550×; Phaant1: 567×; Antarctic multimapping reads: 708×), and *P. cf. pouchetii* was >140× more abundant in the Arctic than in non-polar stations. Additionally, we identified several MAGs significantly associated with polar stations (Mann–Whitney U test), namely:

- TARA_AOS_82_MAG_00183, labeled as *Phaeocystis* sp. 1, branching with *P. antarctica* and *P. cf. pouchetii*; 23.7× more abundant in the Arctic;
- TARA_SOC_28_MAG_00067 and TARA_SOC_28_MAG_00074, labeled as *Phaeocystis* sp. 2 and 3, respectively, branching outside the *P. globosa/antarctica/pouchetii* clade, 78.8× and >1300× more abundant in the Southern Ocean, respectively;
- TARA_SOC_28_MAG_00056 and TARA_ARC_28_MAG_00248, branching in the broader *P. jahnii* clade (within „Phaeo2“), 360× and >77.6× more abundant in the Southern and Arctic Ocean, respectively. Their branching on the timetree suggests that the former speciated ~33.3 Mya, coinciding with the first glaciation of Antarctica 34 Mya, while the latter speciated ~11.8 Mya, largely coinciding with its reglaciation 14 Mya (for interactive tree, see:

<https://itol.embl.de/tree/971244734135541741717954>)

These results suggest that *Phaeocystis* spp. have colonized polar waters multiple times through convergent evolution, although more complete genomes are needed to confirm their identity and assess their specific polar adaptations. Notably, given that *P. jahnii* is widely recognized as a mid-latitude species, the discovery of polar specialists closely related to it is unexpected and raises questions about potential competition with *P. antarctica* and *P. pouchetii*. This underscores the challenges in classifying *Phaeocystis* spp. across diverse environments, owing to their inconspicuous appearance as nanoflagellates, which likely hampers their detection and study in natural ecosystems.

- We newly conclude on I. 227: "We hypothesize that variable rates of mixotrophy and mitochondrial transcription contribute to this flexibility, and affect ecological niche partitioning between *Phaeocystis* lineages."

I agree that this sentence provides clarification.

Response: We appreciate the feedback and consider this remark satisfied.

- To make the narrative more cohesive, we revisited our analysis of horizontal gene transfers in more detail and added the following to Supplementary Results 6: "A total of 183 horizontal gene transfer (HGT) events were recorded, passing our most stringent criteria (see Supplementary Methods 8).

This new paragraph is indeed very interesting from an evolutionary, functional, and ecological perspective.

Response: Thanks very much, we consider this remark satisfied.

- We appreciate the raised concern and acknowledge that statistical analyses of compositional data are prone to biases, particularly in the absence of quantitative standards. For the metatranscriptomic and metagenomic analyses we used TPM normalization, which is commonly used when quantitative standards are not available (Cohen et al., 2022; doi: 10.3389/fmars.2022.867007).

A TPM normalization does indeed allow for normalization by transcript length and the number of sequenced reads, enabling the comparison of gene expression for transcripts of varying lengths across multiple samples. However, this method does not address compositional biases; it maintains them. A compositional bias occurs when the relative proportions of transcripts within a sample influence the results. This type of bias is particularly pronounced in cases where:

- i) A high expression of a few transcripts (e.g., highly abundant genes) dominates the majority of reads.
- ii) Variations in the relative proportions of transcripts between samples lead to distortions in the evaluation of less abundant genes.

If the authors are unable to use methods that avoid these biases or estimate their impact, they should, at a minimum, temper certain conclusions about expression comparisons or explain why potential biases would not affect their conclusions.

Response: We were not focusing on low-abundance functions too much; most of the protein domains listed in Results (lines 234-246) and Supplementary Results 8 are well expressed, and we added additional columns to Supplementary Table S6 to document this (mean TPM and TPM percentile in each biotope, sheet KW test). Rather than differential absolute expression, we were more interested in relatively enriched high-abundance transcripts in each respective biotope, for which TPM normalization coupled with Kruskal-Wallis testing represent a suitable approach.

We chose a more suitable term and replaced “expression” with “enrichment” where appropriate (including Figure 3 legend), and added the following narrative in Supplementary Results S8 to clarify this:

Due to the compositional nature of the expression data and the use of TPM normalization (see Supplementary Methods S4), the results of these Kruskal-Wallis tests should be interpreted as differential enrichment rather than absolute differential expression.

Importantly, as a non-parametric rank-based test, Kruskal-Wallis is robust to deviations from normal data distribution and outliers. Yet, we are indeed open to test our conclusions with alternative approaches. To that end, we employed the tool mentioned by the Reviewer in their first review, ANCOM-BC. Specifically, we used ANCOM-BC2, which allows pairwise comparison of multiple groups, consistent with our experimental design. To better account for different

sequencing depth in microbial load estimates (Tara Oceans libraries are on average 8-10 times more deeply sequenced than our CCE/SOC datasets), we decided to pre-process raw counts by normalizing to sample sequencing depth, and aggregated non-target taxa transcription. We added to Supplementary Methods S4:

Additionally, differential expression was assessed on cluster raw counts normalized to library size (the three biotope sets differ substantially in their sequencing depths; abundances attributed to non-target taxa were aggregated) by ANCOM-BC2 using pairwise comparisons of the three biotopes/groups of interest. We used a prevalence cutoff of 0.05 to avoid the removal of clusters exclusively present in the Southern Ocean biotope (with the smallest sampling size).

ANCOM-BC2 predicted different biomass to Phaeocystales in each biotope's samples, generally corresponding to their library size-normalized raw read proportion (in the whole community). That being said, CCE samples were estimated to have lower biomass/overall expression than polar samples. This seems to directly affect how many clusters/pfams are found overexpressed in CCE:

Compare with TPM/KW (the difference in data point number is due to different data filtering by our TPM/KW implementation and ANCOM-BC2):

We note in Supplementary Results S8:

...the estimated CCE biomass is also consistently lower than in the polar biotopes (average Phaeocystales CCE expression is estimated to be 35.7 and 37.3 % that of Southern Ocean and Arctic, respectively).

We also note that some clusters of interest absent in the temperate data (e.g., PF11999 ice-binding like, PF01906 heavy metal-binding, PF05768 glutaredoxin-like, and PF03203 mercury-resistance protein) were not properly statistically assessed by the algorithm, e.g., while some group2-group3 (Arctic-temperate) pairwise comparisons received a (significant) q-statistic value, the group1-group3 (Southern Ocean-temperate) pairwise comparisons resulted in $q = 1$. All temperate expression values in these examples were estimated as N/A (invalid) and probably represented absent/lowly expressed gene clusters, and as such the difference in group2-group3 and group1-group3 pairwise significance results is a miscalculation. The algorithm manual does not touch on this subject, and it is beyond the scope of this work to assess the extent of its consequences. Due to these limitations, we see TPM/Kruskal-Wallis and ANCOM-BC2 as complementary approaches that can both inform our interpretation of Phaeocystales physiology. Finally, we stress that there is currently no consensus on how to properly address compositional biases, and our initial choice was based on the widespread use of TPM normalization when RNA spike-ins are not available (e.g. Carradec et al. 2018 A global ocean atlas of eukaryotic genes, Nat Comm 9:373 – “reads per kilo base covered per million of mapped reads” RPKM metric; Krinos et al 2023 Reverse engineering environmental metatranscriptomes clarifies best practices for eukaryotic assembly, BMC Bioinformatics 24:74 – TPM metric).

Most importantly regarding our conclusions, the TPM/KW and ANCOM-BC2 approaches essentially show the same picture; an over-representation of oxidative stress mitigation and metal homeostasis in polar data, and an over-representation of proteolytic activities in CCE.

We added to Supplementary Results S8:

[about Southern Ocean and Arctic enrichment] Generally, these patterns are supported by ANCOM-BC2, which estimates absolute differential expression. ...

[about CCE enrichment] These results are only partly supported by ANCOM-BC2 (Supplementary Table S6), possibly because of the overall smaller Phaeocystales biomass of in CCE. Specifically, of the above enriched Pfams, only some were shown differentially overexpressed in CCE relative to polar data by ANCOM-BC2 (trypsin PF00089, peptidase M48 PF01435, scavenger receptor cysteine-rich domain PF00530, RuBisCO small chain PF00101, ATP synthase subunit A, photosystem protein PF00124, cytochrome F PF01333).

We updated Fig. 3c/d to better illustrate the main findings of our differential enrichment/expression analyses. Specifically, in Fig. 3c, we changed the appearance of clusters without a Pfam annotation to triangles to visually distinguish them from annotated clusters; and we replaced Fig. 3d with four smaller ternary plots, each showing the relative enrichment of selected statistically enriched clusters belonging to functional groups “metal responsive”, “oxidative stress”, and “peptidases”, and “unknown function Pfams”.

We left the paragraph in Results I. 234 as is, because these conclusions are supported by both methods (differential enrichment of TPM/KW and differential expression by ANCOM-BC2).

We also addressed the mitochondrial-to-plastid ratio analysis with a similar methodological comparison of ANCOM-BC2 with ASC. We came to similar conclusions but replaced “endocytosis” with a more general term “membrane trafficking” in Results I.224.

- We adjusted the figures and believe they comply with the Nature Publishing Groups guides for preparing figures, including font size (5-7pt).

In the PDF I received, some figures appear blurry when zooming in, for example, Extended 1E, Extended 4A, and Extended 8E.

Response: These panels were placed in the figures in a raster format. We replaced them for vector images and high-resolution raster images where possible and will communicate the problem with the copy editor to ensure print quality.

- We added the following narrative to Supplementary Results 1 to reflect on these problems: "... likely stems from the limitations of short read-only data assembly. Indeed, according to GenomeScope 2.0 estimates, substantial portions of the data represent short repeats (37-69%; not shown). Due to these technical problems and limitations, repetitive regions remain difficult to assemble and quantify in *Phaeocystis* spp., along with recently duplicated genes they might flank. The varying levels of assembly quality introduce biases in coding and non-coding repeats, which is why we did not attempt to quantify gene family expansion in the fragmented assemblies."

Ok

- "The two annotation pipelines employ different intron-aware gene-finding worker algorithms, but fundamentally use a combination of homology-based modeling, ab initio predictions, and RNA/transcriptome evidence methods. Filtering combines different support metrics to select an appropriate single model to represent each locus. "

The authors provide very little technical information. One of the key principles of a publication is the reproducibility of results and data, to the greatest extent possible. Here, I do not see how this could be achieved without knowing the tools used, the specific computational codes, and the options or parameters applied.

Response: We added a new section comprehensively describing the annotation pipeline (Supplementary Methods S3). Software versions and literature were added to Supplementary Table S1 for a quick reference.

- Response: As for the first reference, the dataset does not contain Yaravirus-like or *Pleurochrysis* sp. endemic viruses 1a and 2, since, as we elaborate in Supplementary Results 2., the key phylogenetic markers used in the study (DNAPolB, RNAPa, RNAPb, TFIIS) are missing in *Phaeocystis* endogenous NCLDVs and other Yaravirus-like viruses. The study therefore does not provide a suitable phylogenetic context for our endogenous NCLDVs (the closest relatives from the dataset would be Asfaviridae with a relatively long branch). The second reference focuses on Mimiviridae, endogenous NCLDVs only distantly related to the *Phaeocystis* NCLDVs described in our manuscript. Nevertheless, we updated the phylogeny in Extended Data Fig S8B and added references to these recent works to account for the development in the NCLDV field.

OK

- Response : We developed these preliminary models to address specific questions, i.e., to predict the fluxes/relative fluxes in different conditions and for estimating mitochondrial/plastid flux ratios. These questions did not necessitate

manual curation of the gene-reaction association, which is a time-consuming process, but as a result they do not allow linking fluxes with any gene-related data (e.g. transcriptomic data) or for finding marker genes for a trophic mode of these species, and can lead to erroneous outcomes. Based on our experience with *Cylindrotheca* (Kumar et al., 2024; doi: 10.1126/sciadv.ado2623), curating these models could take up to 3-4 months, with an additional 3-4 weeks to analyze the model predictions and validate them with environmental data. While these results are surely interesting, we would prefer to address them in a future work; perhaps with a specific focus on mixotrophy in nanoflagellates.

OK

- We replaced “three [genomes] with the highest contiguity” with “three with N50 >30,000”, and replaced “fine-tuned” with “lineage-specific”.

OK (but the unit is missing for N50)

- Response: Rephrased to a non-comparative statement:
“The gene repertoires of three annotated *Phaeocystis* reference genomes reflect an adaptation strategy through gene expansion: repetitive elements, horizontal gene transfer, and full-length endogenous virus insertions.”

OK

- Response: That is probably true, but many of the phenomena we analyze (mixotrophy, endogenous viruses, functional gene family expansion) are emergent topics in algal genomics, and we anticipate their importance will grow with more experimental data. This sentence merely concludes the abstract, and we see it as a teaser to raise interest in the rest of the article.

I understand the authors' perspective. However, I find that this sentence, being overly general, undermines the quality of the analyses presented in this manuscript.

Response: Thank you for the encouragement. We have rephrased the abstract almost entirely, see above.

- “While reports on their global biogeography exist (REFs), they are based on amplicons or partial metagenomes, and do not elaborate on gene-level adaptation.”

I see, but 'partial metagenomes' should be replaced with 'partially assembled genomes' or an equivalent term because "partial metagenomes" is ambiguous.

Response: Fixed.

- Response: This is a great question, thank you for raising it. In the updated manuscript, we utilized the same single copy gene normalization analysis to

quantify organellar genome copies. It appears that the number of plastid genome copies per haploid genome is ~10, and the number of mitochondrial genome copies per haploid genome could be between 1-3, but both copy numbers are elevated in higher latitudes (>50 °N/S). The results are presented in Extended Data Fig. 5B and mentioned in the main text following the information regarding a lack of metaG representation of mitochondrial genomes: “with about 1-3 mitochondrial genome copies per haploid genome (Extended Data Fig. 5B)”.

OK, Very interesting.

Response: Thank you for this suggestion, we found this result interesting as well.

- “Since organellar transcripts are not typically analyzed in metaT studies based on polyA-enriched libraries, we also examined if the organelle-mapping reads correlate in NCOG data, where we have complementary polyA and ribosomal RNA-depleted data. For whole genome-mapping read abundance, we saw a significant correlation, with higher abundance recorded for ribo-depleted samples (Spearman’s rho for mitochondrial data ranging 0.71-0.79, except the least abundant *P. rex* with 0.51, and p-values always $<10^{-20}$; Spearman’s rho for plastid data ranging 0.78-0.82 and p-values always $<10^{-60}$; polar genomes were omitted from the analysis). For gene-level mapping abundances, we analyzed *P. globosa* genotype 3 and *P. cordata*. Similarly, most genes were significantly correlated (31 of 38 mitochondrial genes with p-value $<10^{-4}$, mean rho 0.42; 200 of 232 plastid genes with p-value $<10^{-4}$, mean rho 0.47).”

OK. This is interesting.

- Response: Phaeocystis is recognized as its own plankton functional type, separate from other nano- and picoplankton (Buitenhuis et al 2013). Therefore, biomass estimates (MAREDAT) should be comparable with genetic markers from in situ environmental studies identified and annotated as Phaeocystis spp. (i.e., they represent the same plankton component). The take-home message from Fig 1A is the substantial presence of Phaeocystis in marine environments, regardless of the estimates (note that even mean and median biomass estimates differ quite a bit).

OK

- We updated the text to provide information about this resource in the Figure caption: “Interactive tree available via link in Supplementary Results 6.” Additionally, all figure texts were legible when printed.

OK

Reviewer #2 (Remarks to the Author):

Thank you for addressing all our comments.

Reviewer #3 (Remarks to the Author):

Response: No remarks to address from Reviewers #2 and #3.

Finally, we acknowledge the availability of a chromosome-level assembly of *P. globosa* by Chen et al. (DOI: 10.1016/j.isci.2024.110575), which was published several months after our initial submission. While this is an important new genomic resource, we disclaim here that the analyses presented there are not directly comparable to ours due to different software used (for repeat characterization) and poorer taxon sampling (for protein domain enrichment). Specifically, Chen et al. report much lower repeat content and only use two haptophyte genomes for Pfam domain comparison, one of them (*Chrysochromulina tobin*, in the analysis branching sister to *Phaeocystis*) predicted with substantially fewer genes (16,777, compared to 28,138 for *Chrysochromulina parva*, 30,569 for *E. huxleyi* and 32,618 for *P. globosa* CNS00066), which directly affects those results. Because of this discrepancy, we had omitted *C. tobin* from our analyses and inferred our domain enrichments in *Phaeocystis* + PSC with 20 algal species (including *E. huxleyi* and *C. parva*) as genomic background.

Including the chromosome-level assembly of *P. globosa* CNS00066 in our analyses would require extensive manual curation (to mask repetitive regions, identify endogenous viruses) and reconducting the biogeographic and phylogenomic analyses from the very beginning, delaying the publication of this manuscript by a year or longer. With our dataset being comprehensive on taxonomic order level (i.e. Phaeocystales) and more function-oriented through (meta)transcriptomics, we consider the cost of such extensive reanalysis much higher than its benefit. Nevertheless, we reference this work on multiple occasions where comparisons were straightforward, for instance genome size and subspecies taxonomy. We hope the reviewers would understand and agree with this view.